# A CRISPR screen in intestinal epithelial cells identifies novel factors for polarity and apical transport

**Katharina MC Klee[1,2†], Michael W Hess[2†], Michael Lohmüller[3], Sebastian Herzog[3], Kristian Pfaller[2], Thomas Müller[4], Georg F Vogel[1,4*], Lukas A Huber[1]**

[1]Institute of Cell Biology, Medical University of Innsbruck, Innsbruck, Austria; [2]Institute of Histology and Embryology, Medical University of Innsbruck, Innsbruck, Austria; [3]Institute of Developmental Immunology, Medical University of Innsbruck, Innsbruck, Austria; [4]Department of Paediatrics I, Medical University of Innsbruck, Innsbruck, Austria

**Abstract** Epithelial polarization and polarized cargo transport are highly coordinated and interdependent processes. In our search for novel regulators of epithelial polarization and protein secretion, we used a genome-wide CRISPR/Cas9 screen and combined it with an assay based on fluorescence-activated cell sorting (FACS) to measure the secretion of the apical brush-border hydrolase dipeptidyl peptidase 4 (DPP4). In this way, we performed the first CRISPR screen to date in human polarized epithelial cells. Using high-resolution microscopy, we detected polarization defects and mislocalization of DPP4 to late endosomes/lysosomes after knockout of TM9SF4, anoctamin 8, and ARHGAP33, confirming the identification of novel factors for epithelial polarization and apical cargo secretion. Thus, we provide a powerful tool suitable for studying polarization and cargo secretion in epithelial cells. In addition, we provide a dataset that serves as a resource for the study of novel mechanisms for epithelial polarization and polarized transport and facilitates the investigation of novel congenital diseases associated with these processes.

**\*For correspondence:**
georg.vogel@i-med.ac.at

[†]These authors contributed equally to this work

**Competing interest:** The authors declare that no competing interests exist.

## Editor's evaluation

In this work, Klee et al. carried out a genome-wide CRISPR/Cas9-based screen in human intestinal cell line CaCo2 to uncover factors regulating apical localization of a brush border enzyme. Their findings identified dozens of genes and characterized novel players in apical membrane transport including TM9SF4, anocatmin 8, and ARGAP33. This work provides a useful resource for the study of apical polarity and may aid in the understanding of digestive diseases.

## Introduction

Epithelia are highly specialized tissues that line inner and outer surfaces of various organs of metazoans, performing absorption, secretion, and barrier functions. During polarization, epithelial cells assume their characteristic shape by building specialized apical- and basolateral plasma membrane (PM) domains (*Rodriguez-Boulan and Macara, 2014*; *Apodaca et al., 2012*), which are separated by junctional complexes and characterized by a specific composition of lipids and proteins (*Martin-Belmonte et al., 2007*). The asymmetric distribution of polarity complexes and the mutual exclusion of proteins from one domain by proteins from the other domain are critical for the maintenance of apico- basolateral domains at the cell cortex (*Rodriguez-Boulan and Macara, 2014*; *Román-Fernández and Bryant, 2016*). Additionally, tightly orchestrated transport mechanisms and

machineries, as Rab-GTPases, motor proteins, soluble NSF attachment receptor (SNARE)-proteins, and specific adapter proteins, ensure the establishment and maintenance of specialized membrane domains (*Gaisano et al., 1996*; *Low et al., 1996*; *Weimbs et al., 1997*; *Li et al., 2002*).

Defects in polarization and polarized traffic often cause diseases, such as congenital diarrhea and enteropathies (*Thiagarajah et al., 2018*; *Berni Canani et al., 2010*; *Apodaca et al., 2012*). Microvillus inclusion disease (MVID) is an autosomal-recessive enteropathy (*Cutz et al., 1989*), characterized by intractable diarrhea in neonates (*Cutz et al., 1989*; *Ruemmele et al., 2006*). Enterocytes of MVID patients show loss of brush-border microvilli, formation of so-called micro-villus inclusions and subapical accumulation of so-called 'secretory granules' (*Cutz et al., 1989*; *Phillips et al., 2000*). Our studies identified mutations in *MYO5B*, *STX3*, and *STXBP2* to be caus-ative for MVID (*Müller et al., 2008*; *Ruemmele et al., 2010*; *Wiegerinck et al., 2014*; *Vogel et al., 2017b*); they revealed that a molecular transport machinery involving myosin Vb (myo5b), the small Rab-GTPases Rab11a and Rab8a, the t-SNARE syntaxin3 (stx3), and the v-SNAREs slp4a and vamp7 is essential for apical cargo delivery (*Vogel et al., 2015b*; *Vogel et al., 2017b*). This cascade is required for the delivery of apical transmembrane transporters that are important for proper physiological function of enterocytes, such as sodium-hydrogen exchanger 3 (NHE3), glucose transporter 5 (GLUT5), and cystic fibrosis transmembrane conductance receptor (CFTR), but not for dipeptidyl-peptidase-4 (DPP4), sucrase-isomaltase (SI), and amino-peptidase-N (APN). This suggests the presence of additional trafficking routes and transport mechanisms for these apical cargos.

Because the molecular signals for sorting and transport of apical cargo are thought to vary widely, several mechanisms have been proposed to underlie epithelial protein secretion (*Levic and Bagnat, 2021*). A common, characteristic feature of apical cargo is the presence of post-translational modifi-cations, such as N- and O-linked glycosylations that are recognized by specific lectins, as well as GPI-anchors that allow sorting into cholesterol-rich lipid microdomains (*Weisz and Rodriguez-Boulan, 2009*; *Zurzolo and Simons, 2016*). Additionally, recent studies have proposed that protein oligomer-ization coincides with sorting into specialized membrane domains in the *trans*-Golgi network (TGN), which depends on the pH regulation of the TGN lumen (*Levic and Bagnat, 2021*; *Levic et al., 2020*).

To uncover protein functions for a wide range of cellular processes, genome-wide clustered regu-larly interspaced short palindromic repeats (CRISPR)-mediated screens have advanced to a state-of-the-art strategy (*Shalem et al., 2014*; *Shalem et al., 2015*; *Kampmann, 2018*). In addition to their application to understanding the regulation of tumor biology, viral infection, or miRNA processing, CRISPR-mediated screening approaches have recently proven highly effective in discovering novel factors for intracellular protein trafficking and secretion (*He et al., 2021*; *Zhu et al., 2021*; *Hutter et al., 2020*; *Stewart et al., 2017*; *Popa et al., 2020*; *Bassaganyas et al., 2019*). Additionally, the CRISPR-Cas9 technology has been successfully used in Madin–Darby canine kidney (MDCK) cells with the generation of a collection of Rab-GTPase knockouts, which has provided great value for pheno-typic analyses of Rab-KOs in epithelial cells (*Homma et al., 2019*).

In this study, we employed the CRISPR-screening technology as an unbiased experimental strategy to uncover novel regulators of epithelial cell polarization and trafficking by investigating factors required for the apical delivery of DPP4. The brush-border hydrolase DPP4 is a type II transmembrane protein. It is heavily modified with N- and O-linked glycans in its extracellular domain (*Misumi et al., 1992*; *Baricault et al., 1995*; *Fan et al., 1997*), which have been suggested to be critical apical sorting determinants of DPP4 (*Alfalah et al., 2002*). Even though several studies have suggested diverse traf-ficking routes for DPP4, the mechanisms and protein machineries underlying these processes remain enigmatic so far (*Casanova et al., 1991*; *Baricault et al., 1993*; *Low et al., 1992*; *Sobajima et al., 2014*).

Here, we conducted the first CRISPR screen in human intestinal epithelial cells to date. We present an experimental strategy for applying the CRISPR screening system in polarized epithelial cells to study novel protein functions. We have developed an easy-to-use and adaptable, FACS-based assay to measure the efficiency of protein secretion in polarized epithelial cells after genome editing. In combination with a detailed characterization of selected proteins by immunofluorescence and cryo-based electron microscopy, we have identified novel factors required for proper apico-basolateral polarization and secretion of apical cargo. Therefore, our dataset serves as a foundation for future studies aimed at deciphering novel mechanisms underlying epithelial polarization and polarized cargo

transport. In addition, it provides a powerful resource for the investigation and validation of new congenital disease genes to be identified.

## Results

### Development of a genome-wide CRISPR screen to identify factors required for plasma membrane localization of the apical cargo DPP4

We established an unbiased CRISPR-Cas9-loss-of-function screen to define factors involved in surface targeting of the apical model cargo DPP4 in the enterocyte like colon carcinoma cell line, CaCo2 (*Figure 1*). DPP4 is a type 2 transmembrane protein that can be detected with antibodies binding to the extracellular C-terminus of the protein (*Figure 1A*). We made use of this feature to read out the efficiency of endogenous DPP4 surface delivery by fluorescence-activated cell sorting (FACS) in CaCo2 cells after epithelial polarization. Here, we used a period of 18–21 days, during which surface DPP4 signal is significantly increased in the course of cell surface expansion and specialized polarized trafficking processes (*Figure 1B*). In this context, we aimed to define factors required for apical membrane differentiation and cargo trafficking, thereby leading to a strong reduction of DPP4 after surface polarization. First, we generated Cas9-expressing CaCo2 cells and then transduced two biological replicates at a low multiplicity of infection (MOI) (0.2) using the human lentiviral GeCKOv2 CRISPR-library, selecting for successful viral integration with antibiotic treatment with puromycin. We then seeded the infected CaCo2 cultures at high density and allowed the confluent monolayers to further polarize and differentiate for 18 days. Next, polarized cells were detached, stained for endogenous DPP4, and subjected to FACS, separating those cells with only 10% of surface signal left, due to CRISPR targeting (*Figure 1C and D*). To determine the abundance of gRNAs in sorted versus unsorted cell populations, genomic DNA was isolated and read counts were determined by next-generation sequencing. Subsequent analysis using GenePattern and Galaxy analysis tools enabled the identification of 89 gRNAs significantly enriched in the sorted cell population (p<0.05) and represented genes whose downregulation had resulted in reduced DPP4 surface release (*Figure 1D and E*, *Supplementary file 1A and B*).

To exclude the possibility of aberrant effects caused by vacuolar apical compartment (VAC) formation in our screening workflow, we repeated the FACS screening assay with CaCo2 WT cells treated with colchicine (*Gilbert and Rodriguez-Boulan, 1991*). This treatment would induce VAC formation, but we found no change in apical DPP4 signaling (*Figure 1—figure supplement 1A and B*) and therefore concluded that VAC formation should not interfere with the FACS-based screening approach.

### A genome-wide CRISPR screen in polarized enterocytes identifies factors associated with secretory traffic

Next, we wanted to get a comprehensive overview on the gene classes represented in our list of enriched gRNAs. However, automated KEGG pathway and gene enrichment analyses of our results were insufficient. Hence, we manually analyzed the 89 identified genes for common gene ontology (GO) terms and grouped them accordingly. We listed three GO terms from each category (biological process, molecular function, cellular compartment) for each hit, including the most common GO terms captured by the QuickGO -search database, focusing on including GO terms that indicate a role in the secretory pathway (*Supplementary file 1C*).

Our analysis highlighted several genes, with functions related to the organization of the secretory pathway (*Figure 1E and F*, *Figure 2A and B*), including general organization and maintenance of organelles such as the endoplasmic reticulum (ER), the Golgi apparatus, or protein transport at early steps of the secretory pathway (e.g., *KDELR2, RTN2, GOLGA8O*). Further, identified hits were related to protein modification and transport at *cis*- and *trans*-Golgi compartments (*GALNT2, SYS1*), lipid-biosynthesis (*MTMR2, PIP5K1C*), and vesicle fusion and endocytic recycling (*SNAP29, DSCR3*). Two genes identified were associated with ER-plasma membrane (ER-PM) contact sites (*TMEM110, ANO8*). Furthermore, we found several factors required for various aspects of cytoskeletal organization such as actin-filament organization/polymerization (e.g., *MARCKSL1, ARPC4-TTLL3*), cell adhesion (e.g., *ITGA5, FREM3, MPZ*) but also microtubule organizing center (MTOC)/centriole- and cilium assembly and association (e.g., *CCDC61, CCDC42B, C2CD3*). Finally, we found numerous factors with functions related to DNA-templated transcription and cell differentiation (e.g., *ETV7, NKX2-2, ERF*),

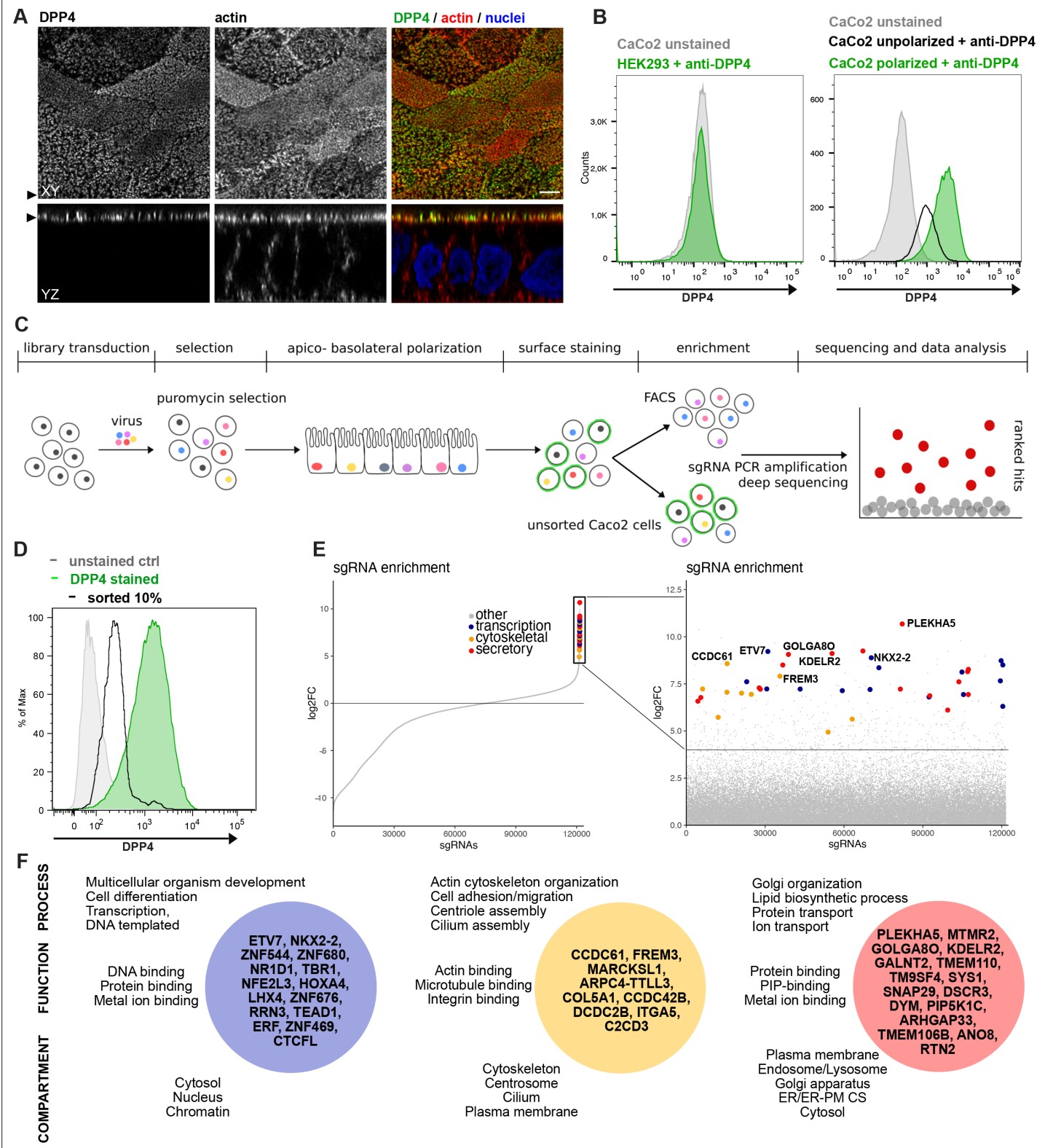

**Figure 1.** A CRISPR-mediated loss-of-function screen in polarized enterocytes. (**A**) Dipeptidylpeptidase 4 (DPP4) localizes to the apical brush-border of polarized enterocytes and can be detected with a specific antibody at its extracellular stalk domain. Top view (XZ) and lateral view (YZ) of a polarized CaCo2 monolayer. Scale = 5 μm. (**B**) During polarization, apical DPP4 is increased due to polarized traffic and surface expansion, which can be measured by flow cytometry (right panel, CaCo2 unpolarized versus polarized). HEK293T cells, not expressing DPP4, serve as quality control for staining specificity.

*Figure 1 continued on next page*

*Figure 1 continued*

(**C**) CaCo2-Cas9 cells are transduced with the lentiGuide-Puro library and selected with puromycin. After selection, CaCo2 cells are seeded to confluent monolayers and cultured for apico-basolateral polarization. Subsequently, cells are detached, stained, and subjected to fluorescence-activated cell sorting (FACS). Sorted and unsorted control cells are processed for gDNA extraction and genomically integrated CRISPR constructs are amplified by PCR. Finally, PCR products of sorted and unsorted cell populations are analyzed by next-generation sequencing and sgRNAs are ranked by their enrichment in the sorted vs. unsorted cell polpulation. (**D**) Sorting was performed for 10% of the cells, with lowest surface-signal intensity, thereby enriching for the cell population that had lost 90% of surface DPP4 signal, due to efficient CRISPR targeting. (**E**) 89 single-guide RNAs were significantly enriched in the sorted cell population. (**F**) Factors enriched in the sorted cell population could functionally be associated with secretory traffic, cytoskeletal architecture, or transcription, in a manual gene -ontology analysis.

The online version of this article includes the following figure supplement(s) for figure 1:

**Figure supplement 1.** Colchine treatment of CaCo2 WT cells, validation of additional gRNA targeting, and complementary ultrastructure and immunoelectron microscopy on the phenotype of selected, polarized KO cells.

as well as mRNA processing/RNA splicing (e.g., *SNRPE, SFSWAP*), translation (e.g., *RPL30, RPL2*), and DNA repair/DNA replication (e.g., *SFR1, ATAD5, REV1*) (*Figure 1E and F*, *Figure 2A*).

Overall, in a CRISPR-mediated loss-of-function screen, we identified a variety of factors that affect surface transmission of an apical model cargo protein, DPP4, at different cellular levels. This underscores the value of our dataset and approach to identify novel factors for secretory membrane trafficking in polarized epithelial cells.

## Novel factors for surface localization of the apical cargo DPP4

After setting up a CRISPR-mediated screening platform in polarized CaCo2 cells, we validated our screening approach by further characterizing potentially novel factors for apical cargo traffic and membrane polarization. Since we had identified several genes with functions related to secretory trafficking (*Figure 2A*), we chose seven promising candidates for further analyses (*Figure 2B*). These factors function on various levels of the endomembrane system: the anoctamin family member anoctamin 8 (ANO8) and the stromal interaction molecule (STIM) enhancing tethering protein STIMATE (TMEM110) are involved in the formation and maintenance of ER-PM contact sites, and in turn, in apical PM-establishment in bile-canaliculi (*Jha et al., 2019*; *Quintana et al., 2015*; *Chun Chung et al., 2020*). The nonaspanin-family member TM9SF4 has been suggested to be required for transmembrane domain sorting in early steps of the secretory pathway but also in the generation of specialized membrane domains in the early *cis*-Golgi compartment (*Perrin et al., 2015*; *Vernay et al., 2018*; *Yamaji et al., 2019*). Polypeptide N-acetylgalactosaminyltransferase 2 (GALNT2) regulates O-linked glycosylation of transmembrane proteins in the Golgi and was chosen as a candidate for screen validation, with a potentially global effect on secretory traffic (*Wandall et al., 1997*; *Moremen et al., 2012*). Sorting nexin 26 (SNX26/ARHGAP33) was included since it has been described as a GTPase-activating protein for Cdc42, a major player in apical domain differentiation (*Kim et al., 2013*). Finally, we chose the lipid kinase subunit phosphatidylinositol 4-phosphate 5-kinase type-1 gamma (PIP5K1C) and the lipid phosphatase myotubularin-related protein 2 (MTMR2) for screen validation and further analysis since they are known regulators of apical phosphatidylinositolphosphate (PIP) pools (*Xu et al., 2019*; *Román-Fernández et al., 2018*).

We generated knockout (KO) cell lines of those candidates using the CRISPR-technology and those gRNAs that had proven to efficiently target in our CRISPR screen (*Figure 3A*). We then analyzed KO cell lines for surface localization of DPP4 by flow cytometry using the previously described polarization assay from our CRISPR screen (*Figure 3B*). These measurements showed that targeting of the selected candidates indeed leads to reduced surface localization of DPP4, but to varying degrees (*Figure 3C*). The strongest effect on DPP4 surface localization was caused by interference with PIP5K1C (~75% reduction), followed by TM9SF4, TMEM110, and GALNT2 (~50% reduction). Interestingly, ANO8-, MTMR2-, and ARHGAP33-KOs showed the mildest phenotype (~30% reduction) (*Figure 3B and C*).

By growing KO cell lines of selected candidate genes and reanalyzing them for the effects of CRISPR targeting on PM localization of DPP4, we validated our primary CRISPR loss-of-function screen and thereby identified new players for surface localization of the apical cargo protein DPP4. For the subsequent analyses of epithelial phenotypes, we generated KO cell lines with a second set of gRNAs targeting the selected seven genes (*Figure 1—figure supplement 1C*) and included these cell lines in the analyses as indicated.

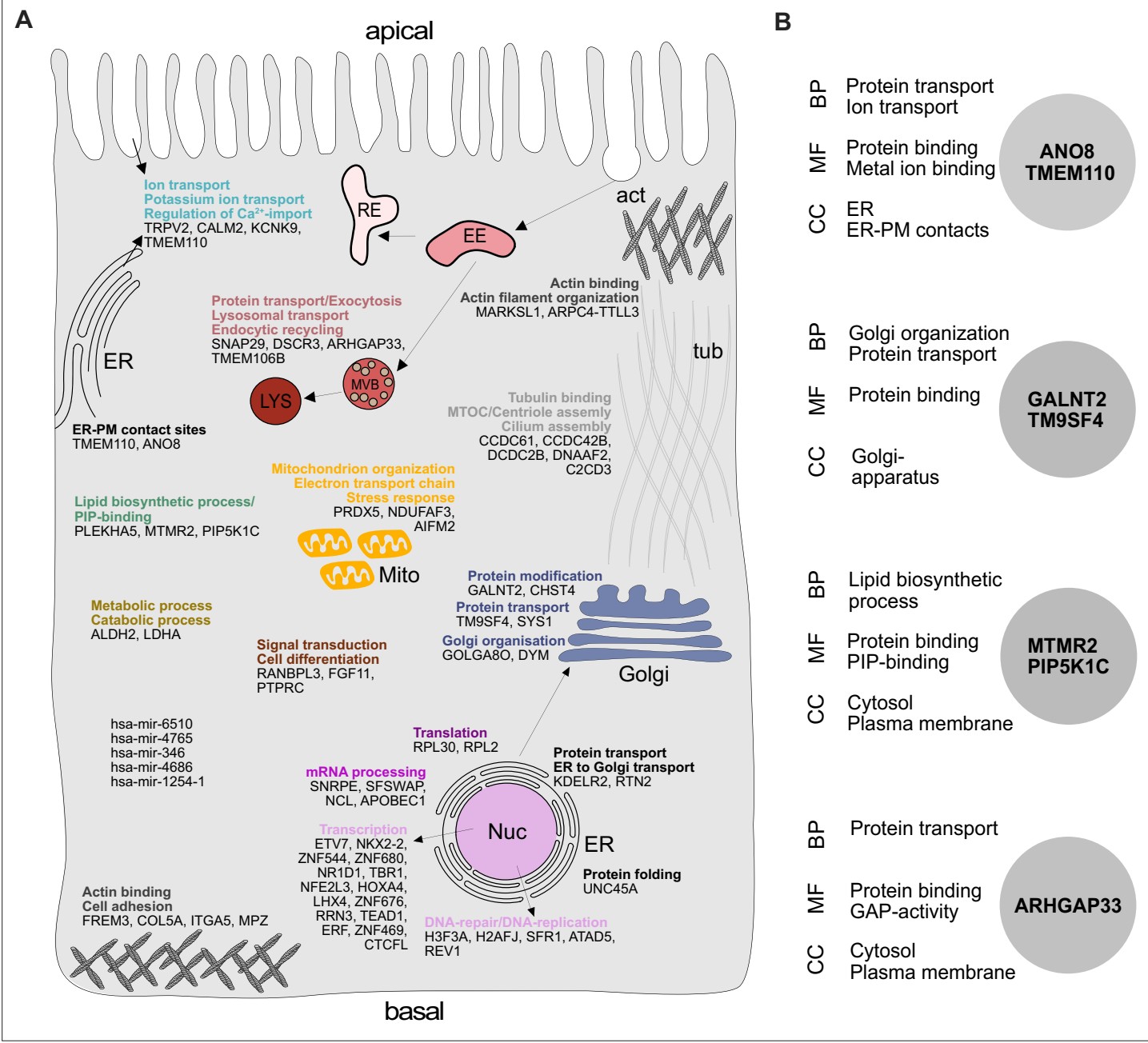

**Figure 2.** Gene ontology (GO) analysis of hits obtained in a CRISPR-mediated loss-of-function screen in polarized CaCo2 cells. (**A**) Schematic representation of significantly enriched genes obtained from a CRISPR screening approach, grouped and organized according to their associated GO terms. (**B**) GO association analysis of the seven factors that were chosen for CRISPR screen readout validation and further characterization. CC, cellular compartment; MF, molecular function; BP, biological process.

## 3D cyst models and high-resolution microscopy reveal novel factors for proper epithelial polarization

Because apical transport and the correct establishment of epithelial polarity are closely linked, we investigated the relevance of the newly identified factors for polarization. Therefore, we performed 3D cyst assays using WT and the corresponding KO cell lines (*Figure 4*). Cysts were analyzed by immunofluorescence microscopy (IF) to determine the targeting of DPP4 to apical membrane domains. We found that all KO cell lines had severe defects in forming a single, central lumen, but rather established multiple or no lumina (*Figure 4A and B*). Although DPP4 was localized in the apical PM domains in all

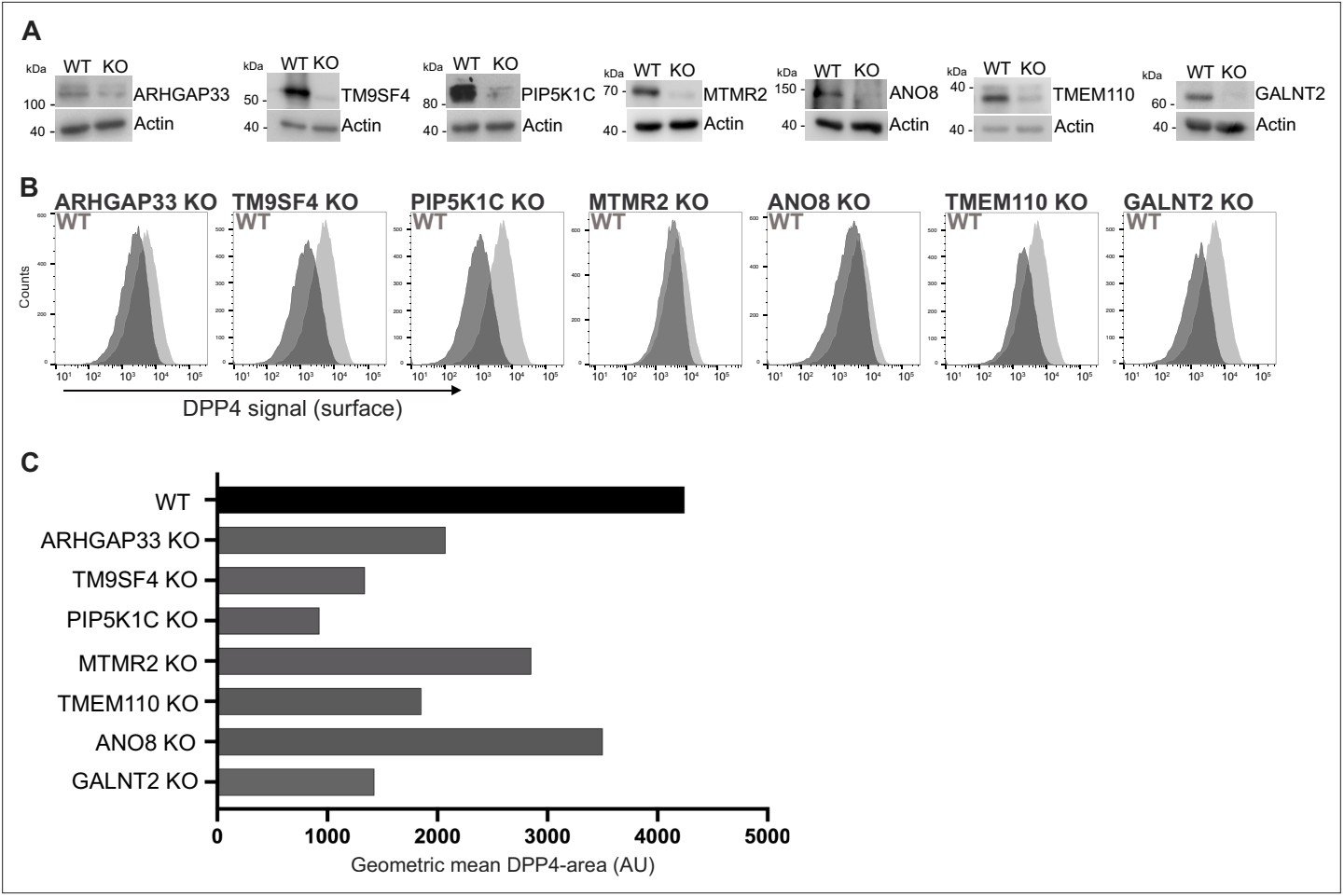

**Figure 3.** Validation of selected candidates identified in a CRISPR-mediated loss-of-function screen. (**A**) Generation of clonal knockout (KO) cell lines of seven candidates chosen for primary CRISPR screen validation and further analysis. Any remaining protein levels in the KO clones after CRISPR targeting were determined by Western blotting compared with wild-type (WT) cells. Beta-actin was used as loading control. Molecular size markers are depicted in kDa. (**B**) The effect of the respective KOs on DPP4 surface transport, in KO cell lines. Cell lines were polarized for 18 days and then subjected to flow cytometry to determine the KO effect on DPP4 surface targeting. (**C**) Geometric means of DPP4-area (DPP4 intensity on the cellular surface) were determined of respective cell lines. All KO cell lines show varying extents of DPP4 surface reduction, with PIP5K1C-KO displaying the strongest and ANO8-KO the mildest effect.

KO cell lines, TM9SF4-, MTMR2-, and ANO8-KO cell lines additionally displayed aberrant intracellular accumulation of DPP4 (*Figure 4A and C*).

To characterize the involvement of the selected candidates in apico-basolateral polarization and apical transport in greater detail, we complemented fluorescence microscopy with cryo-based electron microscopy and investigated the ultrastructural phenotype and the subcellular distribution of selected marker molecules in the respective cell lines. To this end, TM9SF4-, ANO8-, ARHGAP33-, TMEM110-, MTMR2-, PIP5K1C-, and GALNT2-KO cell lines were grown on permeable filter membranes for 18–21 days to obtain fully polarized, differentiated 2D cell monolayers. Samples were then subjected to rapid cryo-fixation (high-pressure freezing and freeze-substitution) for transmission electron microscopy (TEM) or to conventional aldehyde fixation for scanning EM (SEM) and immunogold-TEM.

In contrast to CaCo2 WT cells, all KO cell lines had conspicuous clusters of ectopic microvilli (*Figure 5A–E*, *Figure 1—figure supplement 1D and E*). They appeared either as well-organized ectopic brush-border and adjacent ectopic terminal web, lining distinct intracellular lumina (i.e., typical microvillus inclusions) (*Figure 5A*) or as less complex inclusions filled with intertwined masses of long, curved microvilli (*Figure 5B*, *Figure 1—figure supplement 1E*). Similar configurations occurred basolaterally as paracellular spots lined by densely packed microvilli (*Figure 5C and E*), frequently associated with ectopic tight- and adherens-junctions (*Figure 5E*). In addition, numerous long, curved

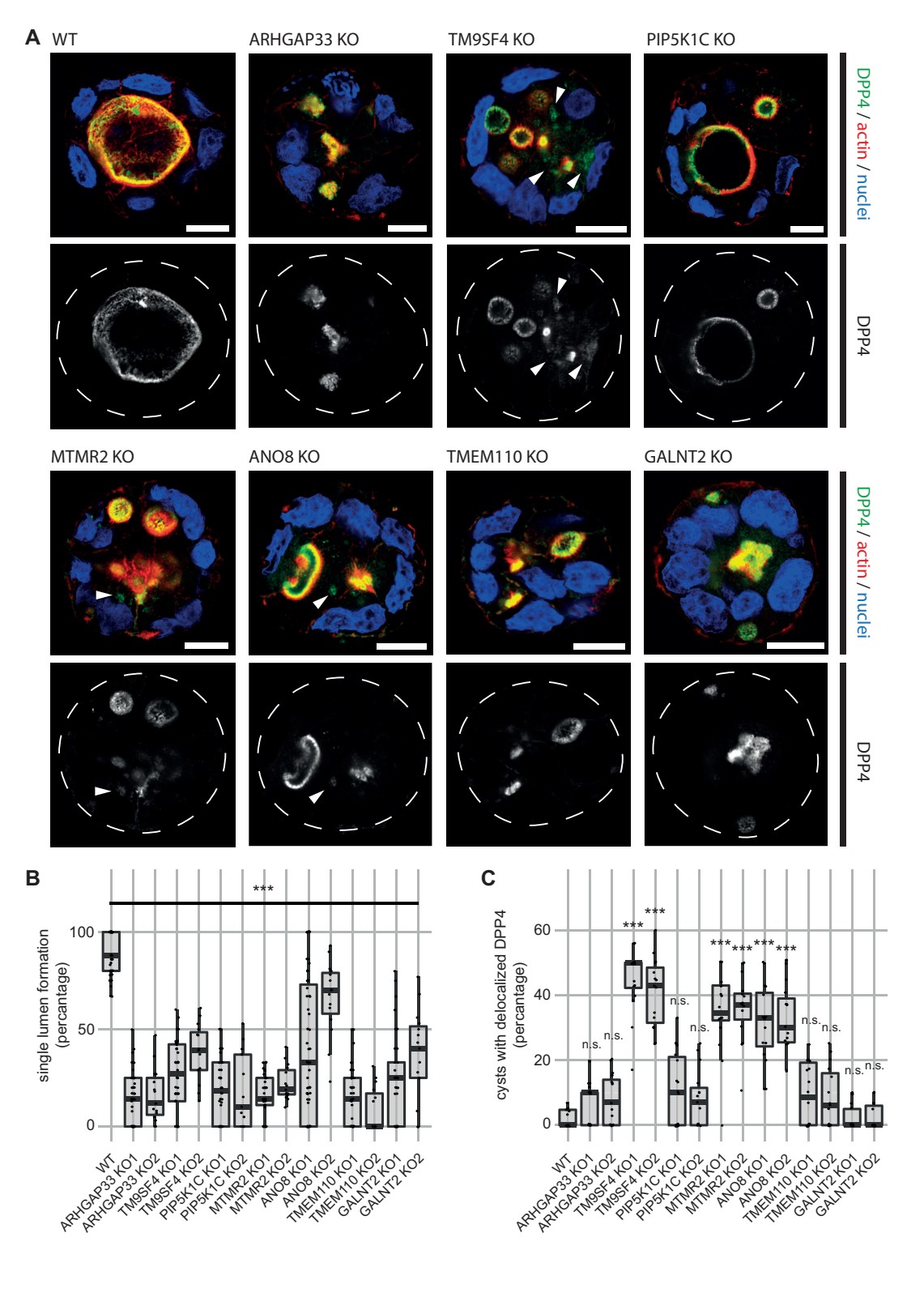

**Figure 4.** 3D cyst cultures demonstrate disrupted epithelial polarity. (**A**) 3D cyst assay were performed with WT and KO cultures. Immunofluorescence micrographs of 3D cysts generated from WT and KO cell lines. All knockdown cell lines form multiple lumina or no lumina. DPP4 localizes to actin-rich structures in al KO cell lines and additionally, to intracellular, actin-negative compartments in TM9SF4-, MTMR2- and ANO8-KO clones (white arrowheads). Scale = 10 μm. (**B**) Single central lumen formation was quantified. The percentage of cysts with a single central lumen is substantially

*Figure 4 continued on next page*

*Figure 4 continued*

decreased in the respective KO cells lines (dot box plot, Mann–Whitney U test. ***p< 0.005, n ≥ 100 cells per condition). (**C**) Delocalized DPP4 in cysts was quantified (dot box plot, Mann–Whitney U test. ***p<0.005, n.s., not significant).

microvilli were found facing the occasionally widened basolateral intercellular space (*Figure 5D*, *Figure 1—figure supplement 1D*). All these patterns (summarized in *Supplementary file 2* and *Supplementary file 3*) together confirm the highly disturbed polarity of all KO cell lines studied here.

Notably, remarkable ultrastructural alterations also involved late endocytic and catabolic organelles (*Figure 6*). In WT CaCo2 cells, the different types and/or maturation stages of multivesicular bodies (MVBs) and lysosomes, as well as some autophagic organelles, appeared normal in size, abundance, and morphology (*Figure 6A and B*) and resembled patterns previously mapped for other cryo-fixed mammalian cell lines (e.g., MEF, HeLa: *Vogel et al., 2015a*; *Yordanov et al., 2019*; *Hess and Huber, 2021*). However, in most KO lines, the late endosomal/lysosomal endomembrane system was characterized – if not even dominated– by partly giant, spherical organelles (in addition to normal MVBs) – at the expense of normal lysosomes (*Figure 5B and E*, *Figure 6C–G*, double arrowheads). These large compartments had either weakly stained, homogeneous granular contents with a few intraluminal elements (*Figure 5B and E*, *Figure 6C–G*), or different amounts of partially degraded material. According to ultrastructural criteria, we tentatively interpreted these poorly structured, faint compartments as types of peculiar endolysosomes or (autophago)lysosomes (*Bright et al., 2016*; *Fujita et al., 1990*; *Remis et al., 2014*). Their size and frequency varied throughout the different KO lines. Especially in TM9SF4- and ANO8-KO, they reached dimensions of up to 2 × 6 µm in diameter (e.g., *Figure 6C*), in other KO cell lines only diameters of approximately 500 nm. In CaCo2 WT cells, we infrequently found this type of organelle as well, but here they had rather normal dimensions (*Figure 6A and B*, double arrowheads; *Supplementary files 2 and 3*).

Regarding the general architecture of the brush-border, SEM and actin fluorescence microscopy revealed more or less severe irregularities in in all KO cell lines studied here (*Figures 7–9*). They included patchy distribution or complete absence of apical microvilli, together with the occurrence of extremely long microvilli (*Figure 7B–H*). We then combined actin labeling with immunofluorescence microscopy using antibodies against the apical components DPP4 and stx3 (*Figure 8A–H*, *Figure 9A–H*). At first glance, we detected DPP4 in most CaCo2-KO monolayers predominantly at the apical plasma membrane. However, more detailed analysis of confocal stacks revealed that DPP4 was also mislocalized to intracellular sites in ARHGAP33-, TM9SF4-, PIP5K1C-, MTMR2-, and ANO8-KO cell lines (*Figure 8B, F, and I*), although to varying degrees. While DPP4 was mislocalized to subapical compartments in ARHGAP33-, TM9SF4-, and ANO8-KO cell lines, PIP5K1C- and MTMR2-KOs displayed DPP4 localization to large, actin-rich, basolateral structures, reminiscent of microvillus inclusions, observed by EM. Consistent with these observations, stx3 was detected at the apical brush-border microvilli in all KO cell lines (*Figure 9B, F, and I*). This was accompanied by additional ectopic localization of stx3 in TM9SF4-, ARHGAP33-, MTMR2-, and PIP5K1C-KOs, with MTMR2- and PIP5K1C-KOs exhibiting stx3-positive, basolateral inclusion-like compartments (*Figure 9D and F*), whereas stx3-positive structures were seen in apical regions in TM9SF4- and ARHGAP33-KO cells (*Figure 9B and E*). We further analyzed the localization of the apical membrane proteins aminopeptiase N (APN) and sucrase-isomaltase (SI) (*Figure 8—figure supplements 1 and 2*). Interestingly, we found APN delocalized to actin-rich intracellular compartments only in ARHGAP33-KO cells (*Figure 8—figure supplement 1E*). However, the apical localization of SI remained unaltered upon deletion of the selected genes (*Figure 8—figure supplement 2A–H*).

To investigate whether KO of each candidate also affects the junctions and differentiation of basolateral domains, we stained all cell lines for the apical tight-junction protein claudin-3, and the basolateral adherens junction protein E-cadherin (*Figure 10A–H*). Our analyses revealed a generally normal distribution of those markers in all KO cell lines. Claudin-3 showed the characteristic localization pattern, with an enrichment towards the apical domain and locally also a lateral membrane distribution and E-cadherin marked basolateral membrane domains (*Figure 10A–H*). Transepithelial electric resistance (TEER) measurements of filter-grown, polarized 2D monolayers of WT and KO cell lines revealed an increase of TEER after measurement day 7 upon KO of ARHGAP33, while TEER of the other KO cell lines remained comparable to those of WT cells (*Figure 10I*).

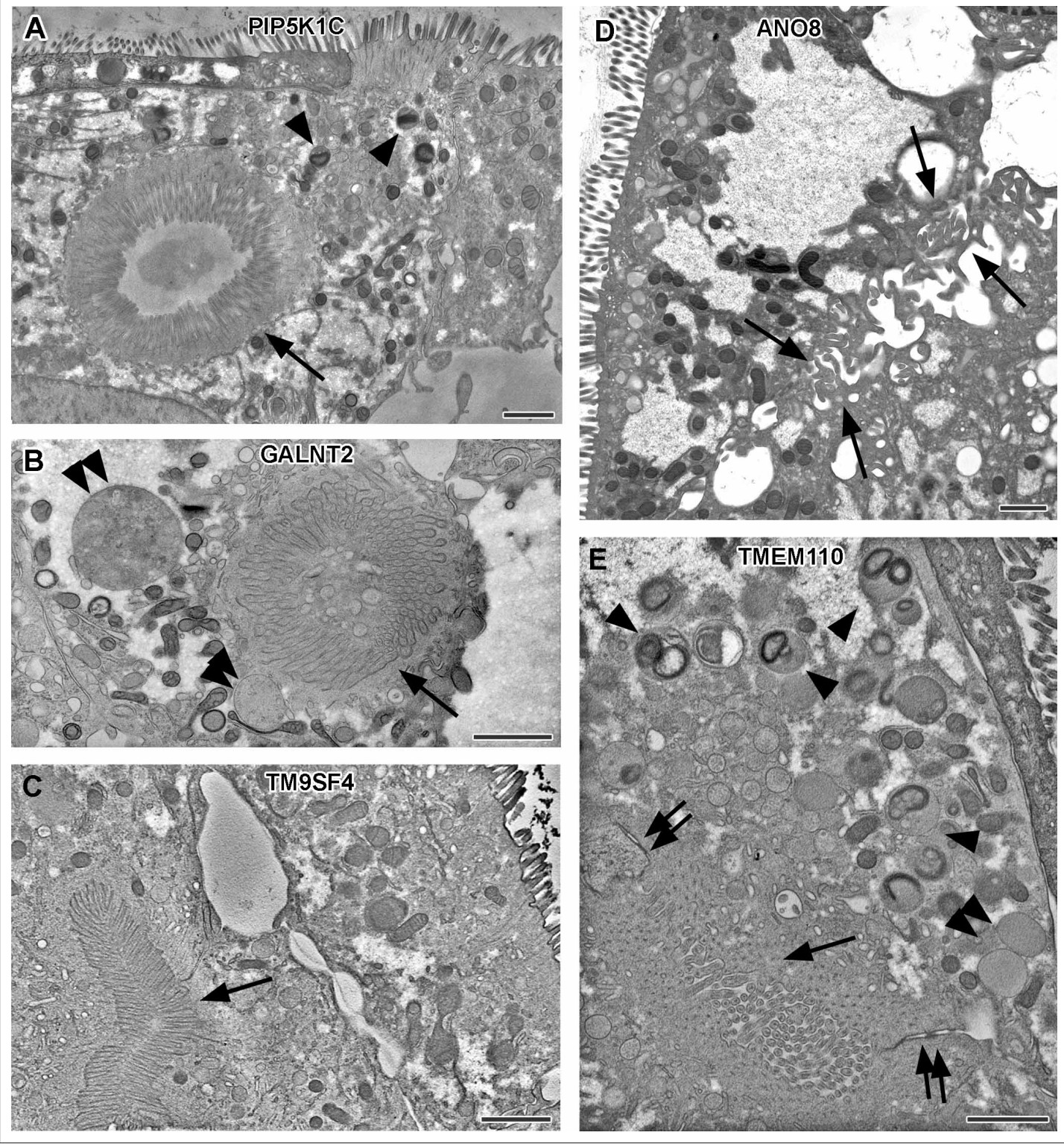

**Figure 5.** Electron micrographs with various configurations of abundant ectopic microvilli reflect polarity defects in 18-day-old 2D cultures of selected CaCo2 knockout (KO) cell lines. (**A**) Distinct intracellular lumen, lined by ectopic brush-border and adjacent terminal web, thus, a typical microvillus inclusion (arrow) inside a PIP5K1C-KO cell. Arrowheads mark lysosomes. (**B**) Spherical intracellular mass of tangled microvilli associated with ectopic terminal web (arrow) inside a GALNT2-KO cell. Double arrowheads mark enlarged late endocytic organelles. (**C**) Paracellular, basolateral spot of densely packed microvilli plus associated ectopic terminal web (arrow) in polarized TM9SF4-KO-cell culture. (**D**) Numerous, slightly bent microvilli facing widened intercellular space (arrows) in polarized ANO8-KO cell culture. (**E**) Paracellular microvillar spot with adjacent ectopic terminal web (arrow)

*Figure 5 continued on next page*

*Figure 5 continued*

and associated cell junctions (double arrows) in polarized TMEM110-KO cell culture. Arrowheads mark lysosomes, double arrowhead marks poorly structured late endocytic organelles. (**A–E**) Scale = 1 µm.

Taken together, our analyses have so far provided several indications of polarity defects after elimination of the selected candidates (Figure 13A and B, *Supplementary file 2*): in a 3D polarization assay, all KO cell lines showed severe defects in forming normal cysts with a single, central lumen but rather generated multiple lumina or no lumina at all. Additionally, TM9SF4-, ANO8-, and MTMR2-KO cell lines showed intracellular mislocalization of the apical marker DPP4 in 3D cysts. TEM of polarized 2D monolayers revealed that KO of all candidates induced ectopic intra- and paracellular clusters of microvilli, reminiscent of typical microvillus inclusions, and SEM complemented these findings with observed alterations in apical brush-border in all generated KO cell lines. Furthermore, TEMs revealed partly extremely enlarged endolysosomal compartments in all KO cell lines except for GALNT2-KO. Notably, the ratio between endolysosomal organelles and canonical lysosomes was partly considerably altered in most KO cell lines at the expense of lysosomes (*Supplementary files 2 and 3*). Finally, immunofluorescence micrographs finally confirmed these observations and additionally indicated partial mislocalization of apical proteins.

## Apical markers mislocalize in enlarged, degradative compartments of TM9SF4-, ARHGAP33-, and ANO8-KO cells

To further complement these results, we used single and double immunogold labeling primarily to characterize the abnormally large, usually poorly structured organelles in TM9SF4, ANO8, and ARHGAP33 KOs and to evaluate their possible association with the abnormal intracellular DPP4 staining in immunofluorescence (*Figure 11A–G*). Membrane or contents of those compartments showed distinct Lamp1, Lamp3, or CathepsinD immunogold label, respectively, in all three KO cell lines (*Figure 11A, C and D*). These findings were also consistent with Lamp1 immunofluorescence micrographs (*Figure 12A–D*). Successful immunogold detection of the previously internalized acidotropic reagent DAMP provided further evidence for the clearly acidic nature of those organelles, justifying their classification as types of modified endolysosomes/(autophago)lysosomes (*Figure 11B*, *Figure 1—figure supplement 1F and G*). They regularly contained mislocalized DPP4 (*Figure 11C and D*, *Figure 1—figure supplement 1F and G*). Moderate, but distinct, aberrant stx3 label that was not detectable by immunofluorescence was also observed in these organelles – in addition to normal apical localization and ectopic localization of DPP4 and stx3 at microvillus inclusions and paracellular microvillar spots (*Figure 1—figure supplement 1L and M*). Further insight into the late endocytic/catabolic endomembrane system of TM9SF4-, ANO8-, and ARHGAP33-KO cells was obtained from starvation experiments. After serum deprivation overnight almost all the giant, poorly structured, faint compartments had disappeared, likely due to autophagic removal, and reformed lysosomes of normal size and morphology were regularly observed (*Figure 1—figure supplement 1H–K*).

Finally, ZO-1 and E-cadherin immunogold labeling allowed us to verify our ultrastructural observations of ectopic tight and adherens junctions associated with paracellular clusters of microvilli that were not seen in the lower resolution immunofluorescence micrographs (*Figure 1—figure supplement 1N and O*).

In summary, our analyses showed that in addition to defects in cell polarization, KOs of all candidates lead to basolateral and/or intracellular mislocalization of apical cargo (*Figure 13A and B*, *Supplementary file 2*). In particular, we observed localization of DPP4 and stx3, to enlarged, endolysosomal/lysosomal compartments, as shown by immunogold labeling of Lamp1, Lamp3, and cathepsin D (*Supplementary file 2*). Moreover, these enlarged compartments were always capable of acidification, and apparently also autophagic degradation and lysosomal reformation. Thus, our observations suggest that KOs of the factors studied, TM9SF4, ANO8, and ARHGAP33, lead to aberrant traffic of apical cargo proteins.

## Discussion

Coordination of molecular polarization and transport machineries in concert with polarized cargo sorting mechanisms is key to epithelial tissue homeostasis. Numerous studies have contributed to our

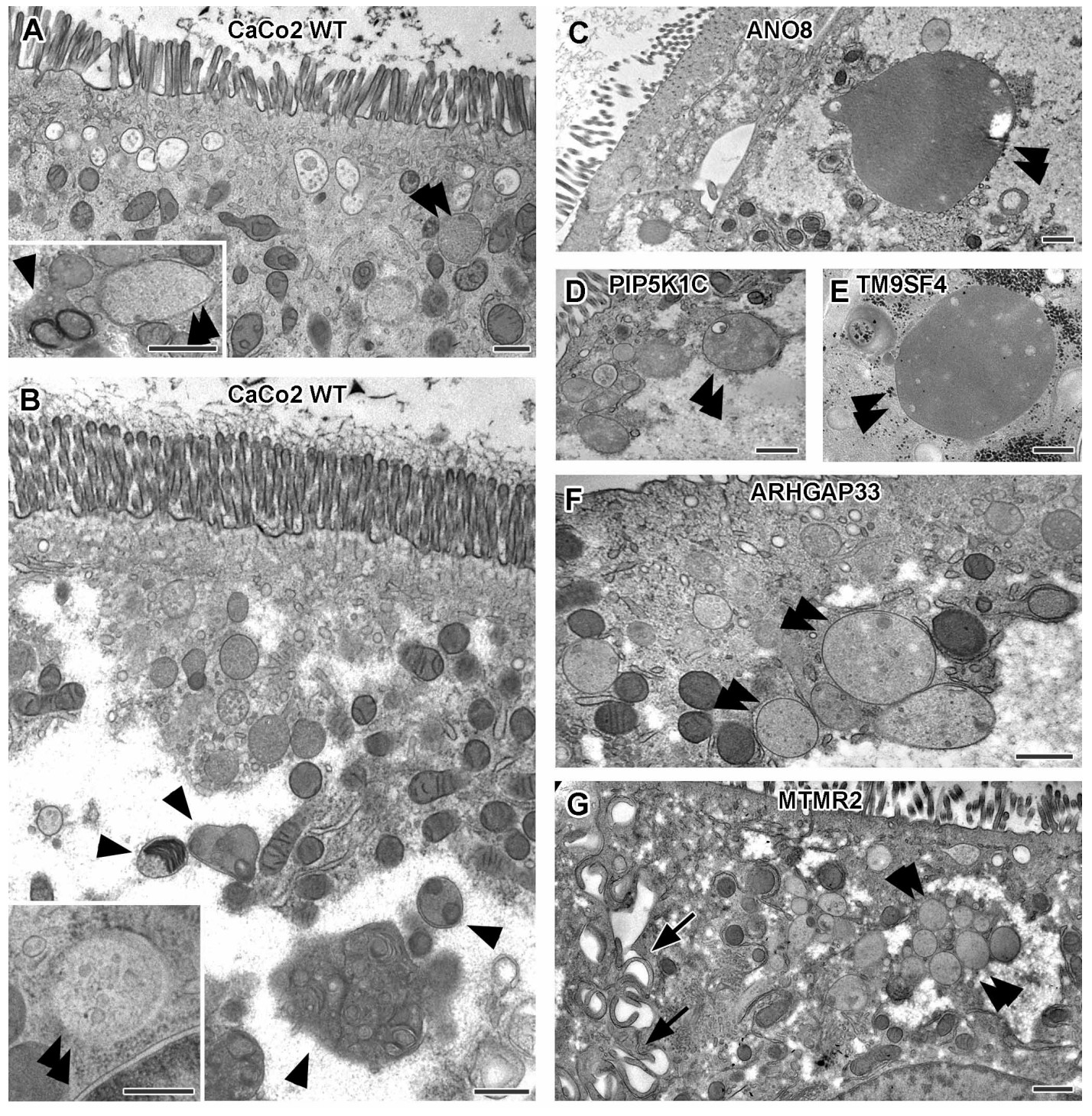

**Figure 6.** Ultrastructure of late endocytic and lysosomal organelles in CaCo2 wildtype (WT) cells versus selected knockout (KO) cell lines of cryo-fixed 18-day-old polarized filter cultures under steady-state conditions. (**A, B**) WT CaCo2 cells with normal endosomes and lysosomes: multivesicular bodies (MVBs) with varying contents (regarding intraluminal vesicle [ILV] number, size, and staining patterns), different types of (autophago)lysosomes (arrowheads), all filled with clearly stained, finely granular material plus membrane remnants (i.e., multilamellar bodies), and/or opaque, amorphous residues (i.e., dense-core lysosomes). Double arrowheads mark examples of inconspicuous spherical organelles with weakly stained, quite homogeneous granular contents harboring only sporadic ILVs and/or other structured components, interpreted as kind of endolysosome (***Bright et al., 2016***). (**C**) Large, poorly structured putative endolysosome (double arrowhead), virtually the predominant type of late endocytic and/or catabolic organelles in ANO8 KO at steady state. (**D**) Moderately sized and enlarged putative endolysosomes (double arrowheads) in PIP5K1C KO cells. (**E**)

*Figure 6 continued on next page*

*Figure 6 continued*

Enlarged putative endolysosome (arrowhead), the predominant type of late endocytic/catabolic organelles in TM9SF4 KO cells. (**F**) Moderately enlarged putative endolysosomes (double arrowheads) occurring together with normal lysosomes in ARHGAP33 KO cells. (**G**) Enlarged putative endolysosomes (double arrowheads) occurring together with slightly smaller lysosomes in MTMR2 KO cells. Arrows mark lateral microvilli. (**A–G**) Scale = 500 nm.

understanding of these processes in the past (*Bryant et al., 2010*; *Levic and Bagnat, 2021*; *Apodaca et al., 2012*). Our previous findings on the mechanisms underlying MVID have elucidated the role of a myo5b-stx3-munc-18-2-dependent trafficking cascade for apical, actin-based cargo delivery; they also suggest the presence of additional trafficking routes and transport mechanisms that direct protein secretion to the apical cortex (*Vogel et al., 2017a*; *Vogel et al., 2015b*; *Vogel et al., 2017b*), which have not been elucidated to date. However, technical advances in CRISPR technology, particularly the development of CRISPR screening strategies, have paved the way for the discovery of protein functions for a wide range of cellular processes (*Popa et al., 2020*; *Hutter et al., 2020*; *Zhu et al., 2021*). In particular, CRISPR-mediated loss-of-function screens have proven to be highly efficient in discovering novel factors for intracellular protein transport and secretory trafficking (*Stewart et al., 2017*; *Bassaganyas et al., 2019*).

Here, we performed the first CRISPR-Cas9 loss-of-function screen in polarized human epithelial cells to identify novel regulators of epithelial polarization and polarized membrane trafficking. We developed a FACS-based assay for the detection of endogenous plasma membrane cargo, which is easy to apply and can be adapted to a variety of transmembrane proteins, given that specific antibodies are available. For our purposes, we used this assay in combination with the highly efficient CRISPR screening system to study genes involved in plasma membrane targeting of the apical model cargo DPP4. Our CRISPR screen identified 89 genes, critically involved in apical targeting of our model cargo, DPP4. This rather moderate number of enriched genes resulted from the high stringency in the screening assay, namely sorting for cells with a quite drastic reduction of surface DPP4 (90%). Even though this allowed to enrich cells with a high gRNA targeting efficiency and increased the specificity of our screen, we thereby also limited our approach in terms of quantity and diversity of the identified hits.

Our experimental approach, combined with GO analysis of the 89 hits, highlighted several genes with functions associated to the secretory pathway. To demonstrate the validity of our dataset, we selected seven factors for phenotypic and morphological characterization, focusing mainly on organelles associated with protein transport.

We demonstrated that the KO of all selected candidates causes disturbed epithelial polarization. This was demonstrated by 3D cyst assays and EM analyses of filter-grown, polarized 2D monolayers, where we detected the formation of ectopic intracellular and paracellular clusters of microvilli. This phenotype was particularly pronounced in KO cells of PIP5K1C, MTMR2, TM9SF4, ANO8, and ARHGAP33, where the localization of the apical components DPP4, stx3 in the intracellular and paracellular microvillar clusters indicated the formation of ectopic neo-/pseudo-apical domains. This highlights yet uncharacterized regulators for epithelial polarization and proposes potentially novel mechanisms for this process. PIP5K1C and MTMR2 are involved in the regulation of PIP pools accounting for apical PM composition. Therefore, imbalances in cellular PIP-pools might be the basis for the observed phenotypes. Interestingly, the mRNAs of MTMR2 and PIP5K1C mRNAs were shown to be expressed predominantly in polarized 2D cultures, whereas they were downregulated in 3D cysts of MDCK cells, indicating differential PIP regulation in 2D polarized monolayers and polarization 'de novo' (*Román-Fernández et al., 2018*). However, our data also hint at a role of enzymes in polarization of 3D cultures, as we observed aberrant lumen formation in KO cells.

Furthermore, the diversity of signals and determinants that coordinate the formation of specialized membrane domains is illustrated by the different functions with which the candidates selected here are associated. The ER-PM contact site proteins ANO8 and TMEM110 might regulate polarization via the control of Ca2+ influx and signaling (*Jha et al., 2019*; *Quintana et al., 2015*), while TM9SF4, which has been implicated in the regulation of glycolipids in the Golgi apparatus (*Perrin et al., 2015*) and VH-ATPase assembly (*Lozupone et al., 2015*), possibly controls polarization through generation of lipid microdomains and pH regulation. GALNT2 might contribute to establishing polarity via its role as O-glycosylating enzyme in the Golgi apparatus and ARHGAP33 as a GAP-protein for the small

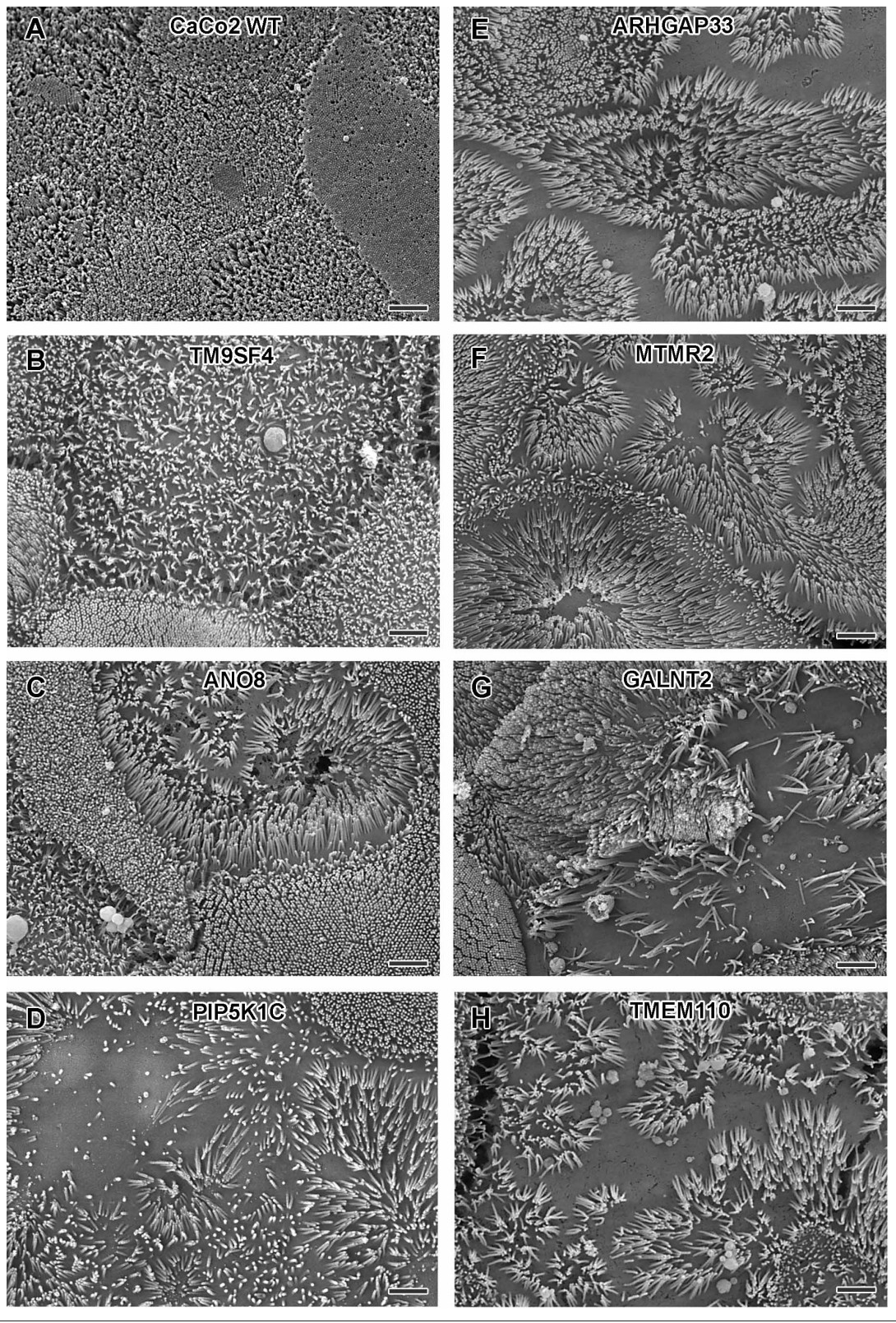

**Figure 7.** Scanning electron microscopy (SEM) surface views on apical microvilli of 18-day-old polarized CaCo2 wildtype (WT) cells versus knockout (KO) cells. (**A**) CaCo2 WT cells with dense, quite uniform brush-border. (**B–H**) Patchy distribution of partly abnormal microvilli characterize the apical surface of all KO cell lines specified here. (**A–H**) Scale = 2 μm.

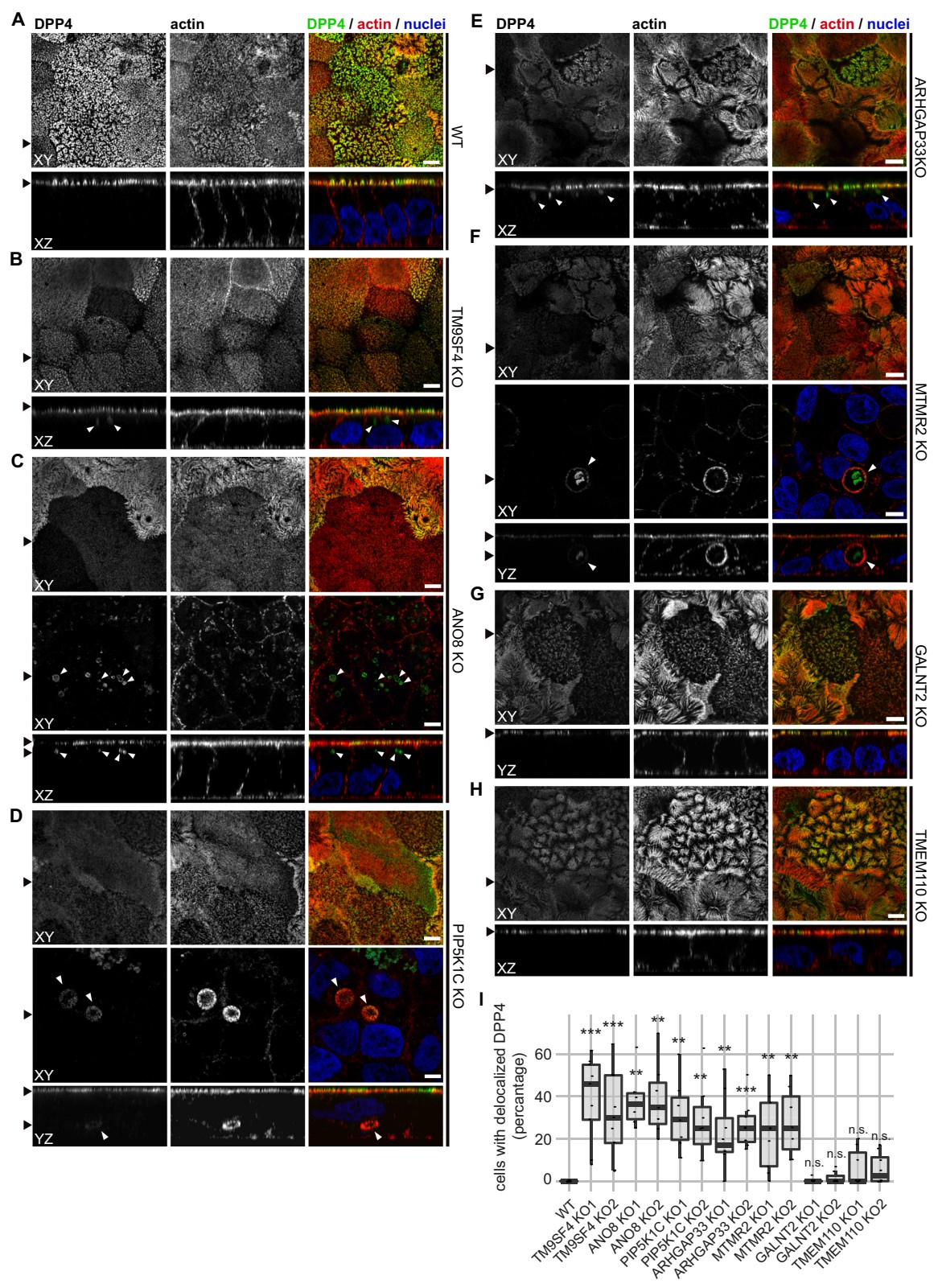

**Figure 8.** Confocal micrographs of DPP4 and actin immunofluorescence staining from wildtype (WT) and respective knockout (KO) cell lines. (**A**) DPP4 localization is restricted to the actin-rich microvillus brush-border in WT cells. (**B, C**) DPP4 can still be targeted to the apical plasma membrane, but also mislocalizes to subapical compartments in TM9SF4- (**B**) and ANO8-KO (**C**) cells (white arrowheads). (**D**) PIP5K1C-KO cell lines display large, DPP4- and actin-positive, basolateral compartments (white arrowheads). (**E**) DPP4 mislocalizes to subapical compartments in ARHGAP33-KO cells (white

*Figure 8 continued on next page*

*Figure 8 continued*

arrowheads). (**F**) MTMR2-KO cell lines display large, actin-rich basolateral compartments, that also show DPP4 (white arrowheads). (**G, H**) In GALNT2- and TMEM110-KO cells, DPP4 localizes strictly to the apical plasma membrane. (**I**) Delocalized DPP4 in polarized CaCo2 cells was quantified (dot box plot, Mann–Whitney U test. **p<0.01, ***p<0.005, n.s., not significant, n ≥ 100 cells per condition). XY = top view of polarized monolayer; XZ/YZ = lateral view of polarized monolayer. Scale = 5 μm.

The online version of this article includes the following figure supplement(s) for figure 8:

**Figure supplement 1.** Confocal micrographs of aminopeptidase N (APN) and actin immunofluorescence staining from wildtype (WT) and respective knockout (KO) cell lines.

**Figure supplement 2.** Confocal micrographs of surcase-isomaltase (SI) and actin immunofluorescence staining from wildtype (WT) and respective knockout (KO) cell lines.

GTPase Cdc42 might itself be critically involved in polarization and polarized traffic as well (*Nakazawa et al., 2016*).

Features indicating disturbed polarization were accompanied by the formation of conspicuous, enlarged Lamp1, Lamp3, and cathepsin D-positive endolysosomal/lysosomal structures, upon KO of all cell lines, that were additionally positive for DPP4 and stx3 in ANO8-, ARHGAP33-, and TM9SF4-KOs. Notably, these compartments showed functional acidification and could undergo autophagic degradation followed by lysosomal reformation. Therefore, we propose that defective polarization upon disruption of one of these genes may be associated with altered cargo transport and/or sorting of apical cargo into lysosomal compartments by various cellular mechanisms. Little is known about the potential role for ARHGAP33 in epithelial polarization; however, one could speculate that a role in modulating Cdc42 might account for the observed polarity defect and the mislocalization/mistrafficking of apical cargo to late-endosomes/lysosomes (*Nakazawa et al., 2016*; *Schuster et al., 2015*). The Golgi apparatus is believed to be a major hub for sorting events of secreted cargo proteins and many signals and mechanisms have been proposed to be major Golgi-associated sorting determinants (*Weisz and Rodriguez-Boulan, 2009*; *Rodriguez-Boulan et al., 2005*). Ca2+ levels in the Golgi apparatus, for example, were shown to regulate apical sorting of GPI-anchored proteins in polarized epithelial cells (*Lebreton et al., 2021*). Thus, mislocalization of DPP4 in ANO8-KOs to endosomal/lysosomal compartments could result from aberrant, Ca2+-dependent sorting in the Golgi apparatus. Finally, lipid microdomains and pH regulation represent major sorting determinants of apical cargo in the Golgi (*Hallermann, 2014*; *Medina et al., 2015*; *de Araujo et al., 2017*; *Schuck and Simons, 2004*). Therefore, it is plausible to assume that the observed mistargeting of apical cargo into lysosomal compartments may coincide with defects in glycosphingolipid synthesis and/or V-ATPase-mediated pH regulation, caused by TM9SF4-KO (*Lozupone et al., 2015*; *Levic et al., 2020*; *Levic and Bagnat, 2021*).

It seems noteworthy that cellular Ca2+ homeostasis, the regulation of intracellular pH, as well as the synthesis PIP species and lipid microdomains regulate a variety of processes related to endocytic recycling of membrane cargo, autophagy, and lysosomal biogenesis (*Medina et al., 2015*; *Sbano et al., 2017*; *Hallermann, 2014*). Defects in either of these processes could therefore be responsible for the observed lysosome-related phenotypes, leading either primarily or secondarily to defective epithelial polarization/secretory traffic.

Because several diseases characterized in the past have been associated with defects in polarized trafficking and protein missorting, we also screened our dataset for all possible genes that have been associated with congenital enteropathies. Apart from the association of Unc-45 Myosin Chaperone A (UNC45A) with syndromal diarrhea and cholestasis, no other genes identified in our CRISPR screen have been published in this context to date (*Esteve et al., 2018*; *Li et al., 2022*; *Dulcaux-Loras et al., 2022*). However, MVID caused by mutations of myo5b, stx3, stxbp2, or unc45a is a prominent example for pathological accumulation of considerably enlarged autophagosomal and/or lysosomal organelles in the cell periphery. Despite their abundance, those catabolic organelles apparently suffer from some degree of overload due to their inability to remove misdirected excess cargo (as reflected by 'secretory granule' accumulation) and to efficiently degrade the ectopic apical domains/microvillar structures. Moreover, abnormal late endosomes/lysosomes with close resemblance to the respective faint, poorly structured lysosomal compartments in the phenotypes we describe here were implicated in another neonatal intestinal disorder, namely, human mucolipidosis type IV. In newly born mice, the absence of mucolipin-1 and -3 induced aberrant swelling of those organelles in enterocytes,

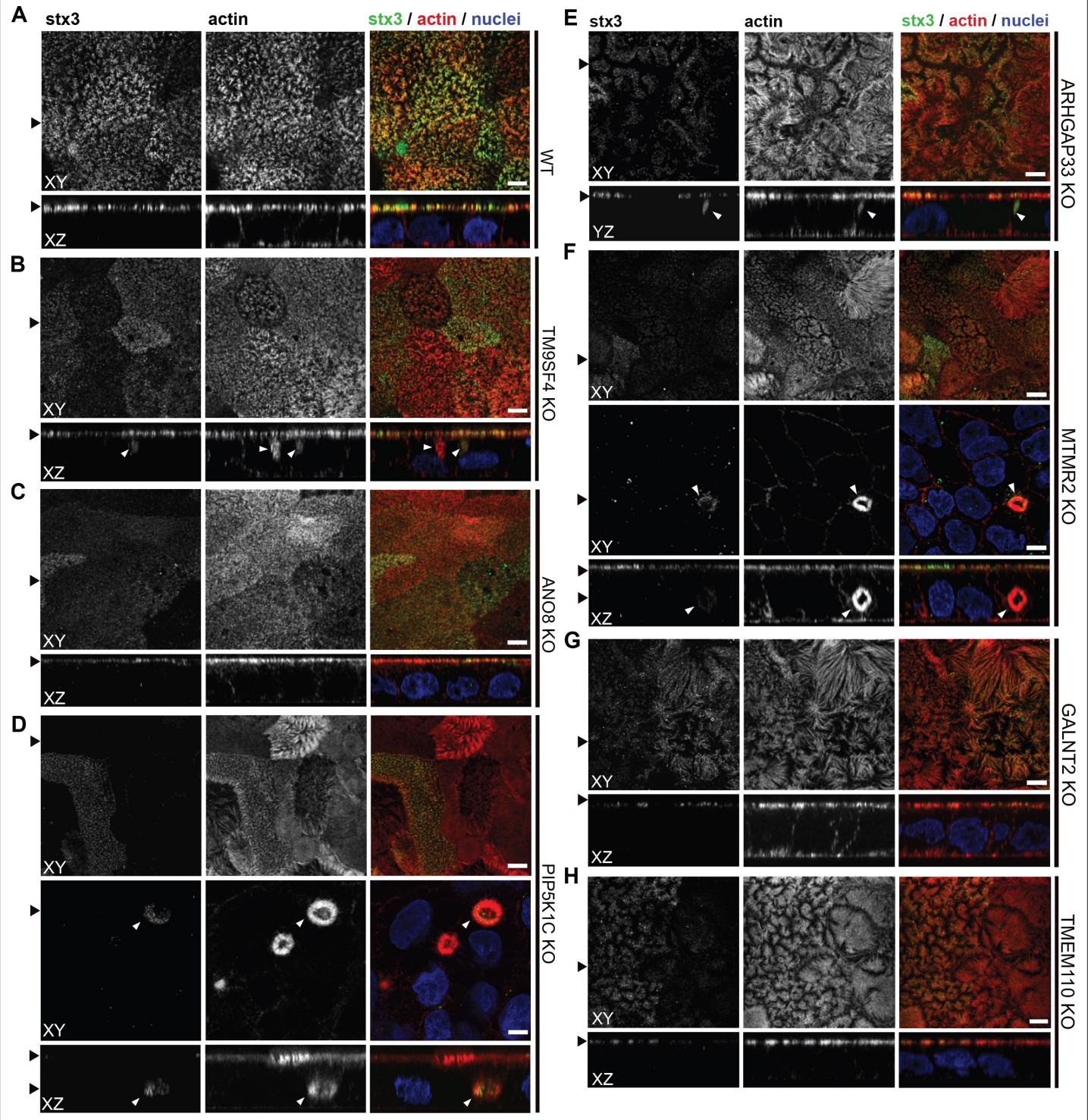

**Figure 9.** Confocal micrographs of syntaxin-3 (stx3) and actin immunofluorescence staining from wildtype (WT) and respective knockout (KO) cell lines. (**A**) stx3 localizes strictly to the apical plasma membrane in WT cells. (**B**) stx3 abberantly localizes to a subapical compartment in TM9SF4-KO cells. (**C**) ANO8-KO cell lines display apical localization of stx3. (**D**) stx3 mislocalizes to large, basolateral, actin-rich compartments in PIP5K1C-KO cells, reminiscent of microvillus inclusions. (**E**) stx3 localizes to subapical compartments in ARHGAP33-KO cells. (**F**) stx3 mislocalizes to large, basolateral actin-rich compartments in MTMR2-KO cells. (**G, H**) GALNT2- and TMEM110-KO cell lines display apical localization of stx3. XY = top view of polarized monolayer; XZ/YZ = lateral view of polarized monolayer. (**I**) Delocalized stx3 in polarized CaCo2 cells was quantified (dot box plot, Mann–Whitney U test. **p<0.01, ***p<0.005, n.s., not significant, n ≥ 100 cells per condition). XY = top view of polarized monolayer; XZ/YZ = lateral view of polarized monolayer. Scale = 5 µm.

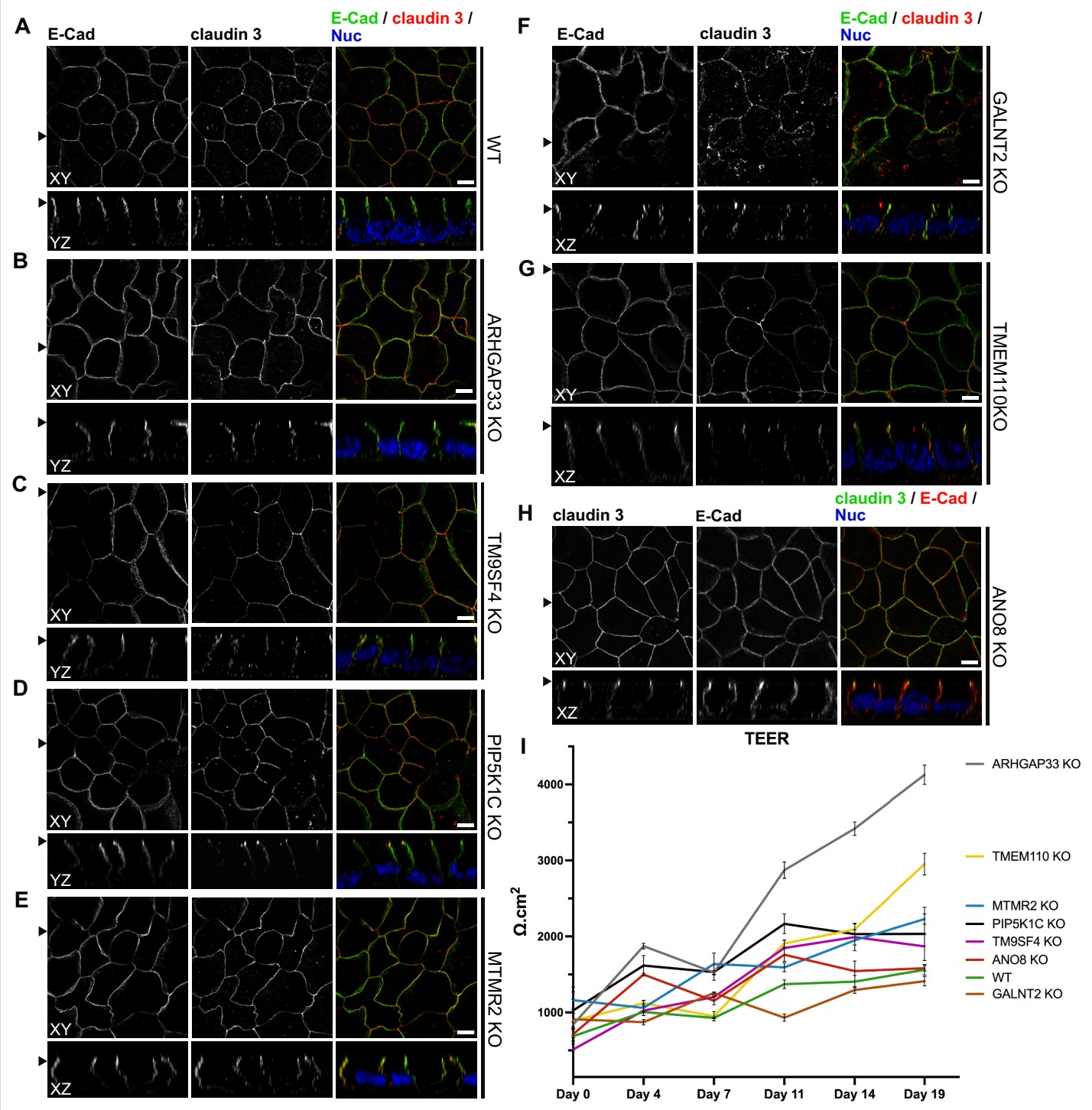

**Figure 10.** Confocal micrographs of E-cadherin and claudin 3 immunofluorescence staining from wildtype (WT) and respective knockout (KO) cell lines.
(**A**) Localization of E-cadherin and claudin 3 in WT cells, where E-cadherin is distributed over the basolateral plasma membrane and claudin 3 is enriched at apical domains, but can also be found at basolateral membrane regions. (**B–H**) ARHGAP33-, TM9SF4-, PIP5K1C-, MTMR2-, GALNT2-, TMEM110-, and ANO8-KO cell lines show basolateral E-cadherin localization as well as claudin 3 enrichment at apical and lateral domains, similar to WT cells (**A**). Scale = 5 μm. (**I**) TEER measurements of WT and respective KO clones. TEER of ARHGAP33-KO cells increases substantially around day 7 of the measurement. Measurements are depicted as means with standard deviation. XY = top view of polarized monolayer; XZ/YZ = lateral view of polarized monolayer; scale = 5 μm.

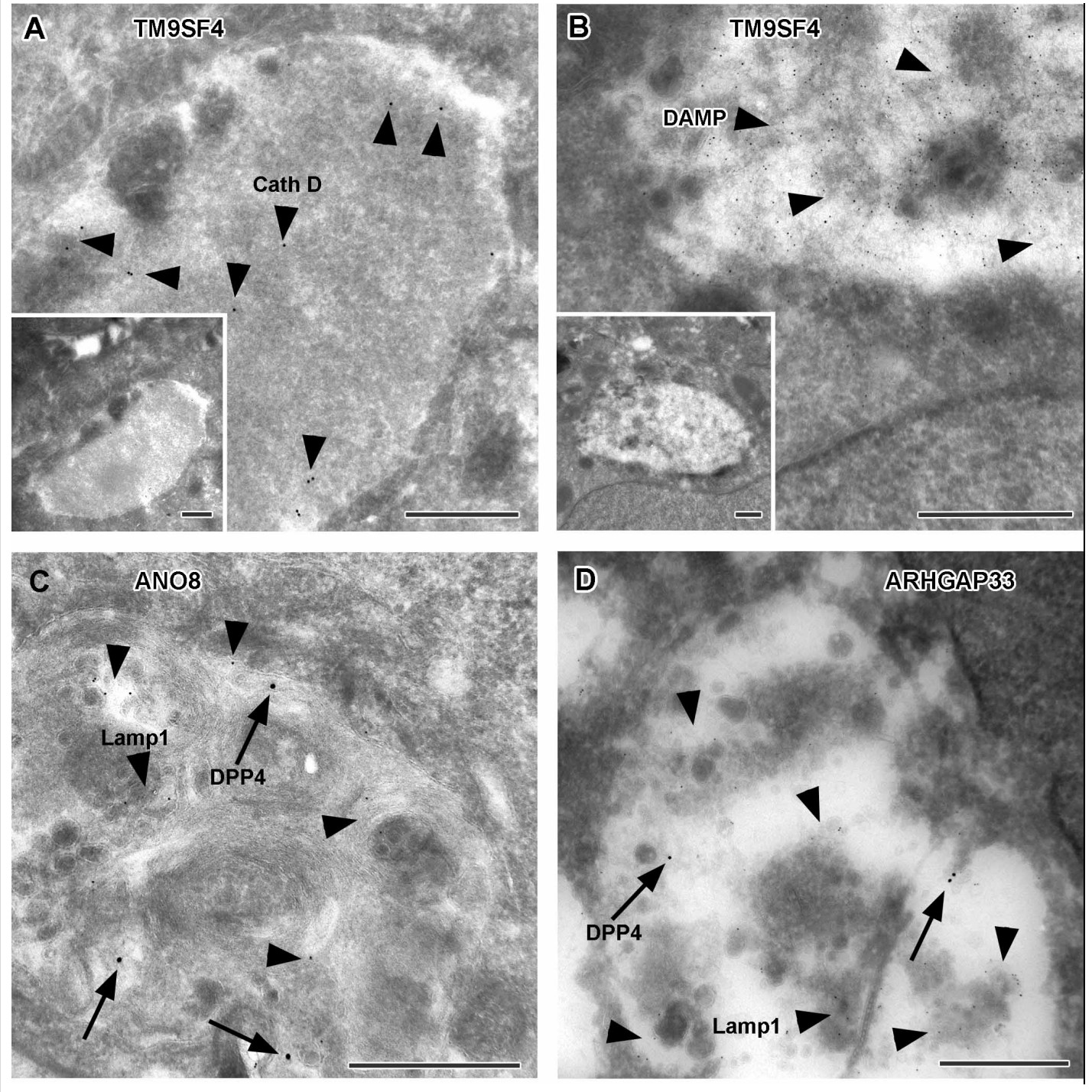

**Figure 11.** Immunoelectron microscopy of late endocytic and catabolic organelles in selected CaCo2 knockout (KO) cells. (**A, B**) Enlarged, poorly structured endolysosomes in TM9SF4 KO cells showing distinct cathepsin D and DAMP-immunogold label (arrowheads) indicative of their acidic contents. Inserts show respective overviews of the depicted organelles. (**C, D**) Mislocalized DPP4 (arrows) colocalizing with LAMP1 (arrowheads) in enlarged compartments in ANO8 and ARHGAP33-KO cells. (**A–D**) Scale = 500 nm.

diminished apical endocytosis from the intestinal lumen, caused diarrhea, and delayed growth (*Remis et al., 2014*). In our opinion, all those examples underline the crucial role of proper establishment and maturation of the highly complex system of (late) endosomal and lysosomal organelles in the small intestine; for example, during the early neonatal period of mammals, especially the transient,

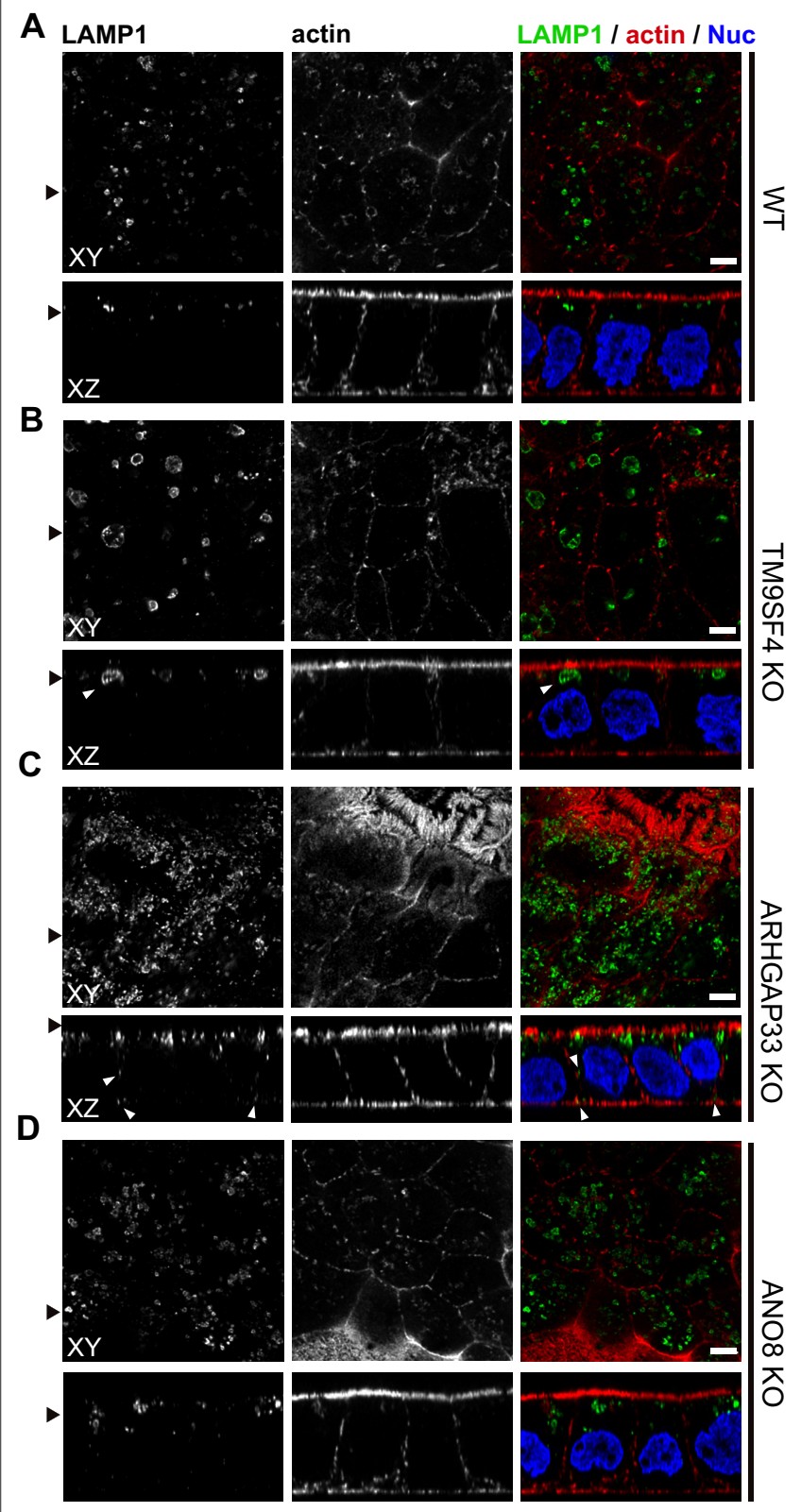

**Figure 12.** Confocal micrographs of Lamp1 immunofluorescence staining from wildtype (WT) and respective knockout (KO) cell lines. (**A–D**) The Lamp1-positive compartments appear enlarged upon KO of TM9SF4 (**B**) and localize to basolateral regions upon KO of ARHGAP33 (**C**) (white arrowheads). Scale = 5 µm.

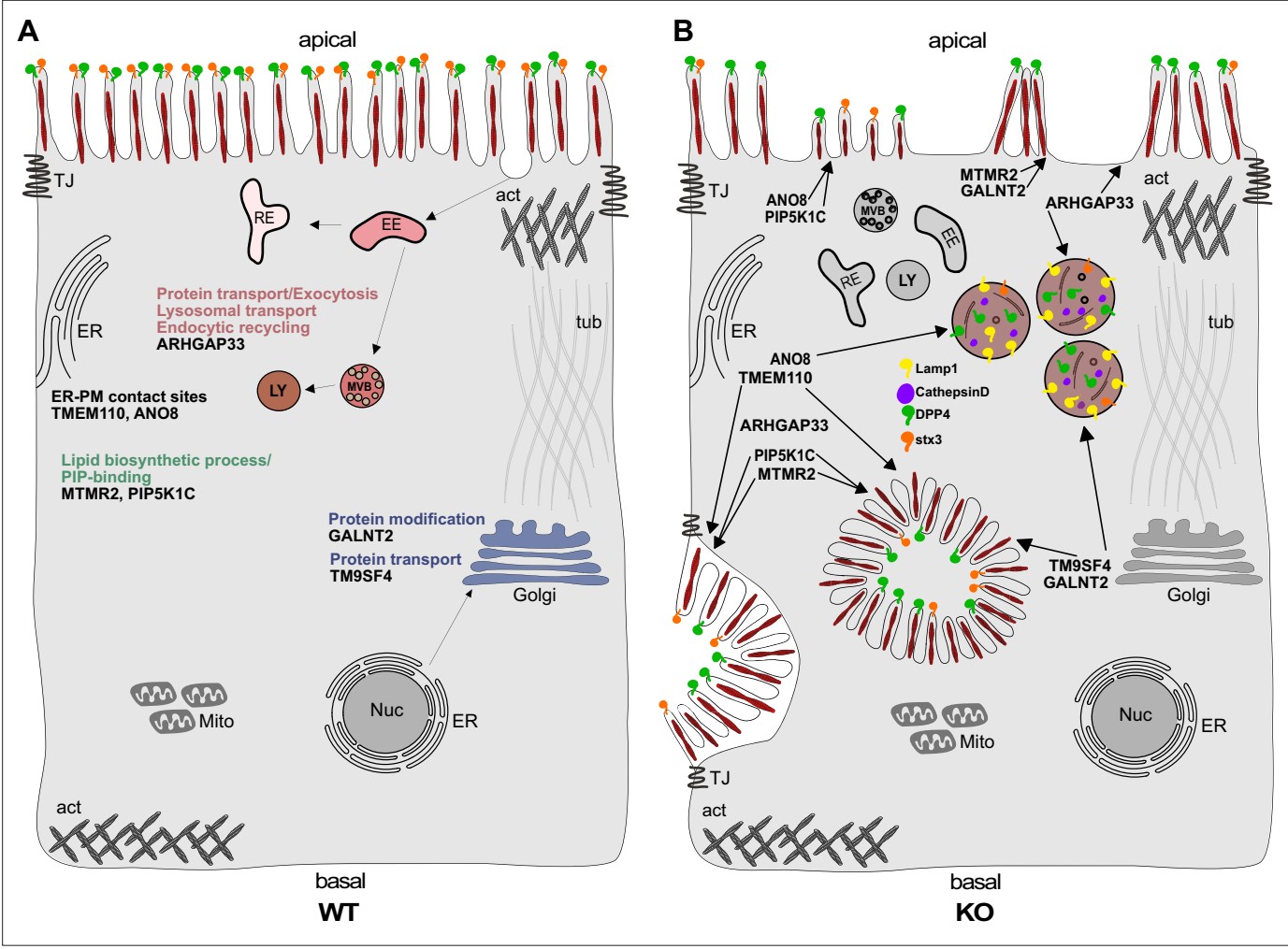

**Figure 13.** Simplified scheme of the phenotypes observed upon knockout (KO) of ARHGAP33, TM9SF4, PIP5K1C, MTMR2, GALNT2, ANO8, and TMEM110. (**A**) Scheme of a healthy enterocyte. The investigated factors for screen validation and further phenotypic characterization are displayed together with their associated gene ontology (GO) terms. (**B**) KOs of ARHGAP33, TM9SF4, PIP5K1C, MTMR2, GALNT2, ANO8, and TMEM110 lead to the formation of microvillus inclusions and lateral pseudo-apical domains with microvilli. KO of ARHGAP33, TM9SF4, and ANO8 leads to the formation of enlarged late endosomal/lysosomal compartments positive for Lamp1 and cathepsin D and that contain the apical markers DPP4 and stx3. All KOs additionally lead to aberrant, 'tipi-like' assemblies of apical microvilli.

so-called 'giant lysosomes' during the suckling period (*Wilson et al., 1991*; *Fujita et al., 1990*; *Remis et al., 2014*). In addition to neonates, this seemingly broad spectrum of overload and/or accumulation of enlarged catabolic organelles with enhanced – but insufficient– autophagy and lysosomal inefficiency has also been described from intestinal disorders of adults, e.g., in patients with necrotizing enterocolitis (*Yamoto et al., 2020*).

Finally, it should be noted that several genes identified in our screening have been associated with neuropathies or myopathies (Charcot–Marie–Tooth syndrome [MTMR2]; Dyggve–Melchior–Clausen syndrome [DYM]) (*Denais et al., 2011*; *Wang et al., 2019*). Thus, it would be worthwhile to examine our dataset in the context of other cellular systems as defective secretory transport and cellular polarization provide a mechanistic basis for a spectrum of pathologies in many tissues and organs.

In summary, this genome-wide CRIPR/Cas9 screen, together with the extensively described and illustrated representation of cellular organelle pathologies, provides a very valuable resource for future investigations aimed at unraveling the complexity and diversity of mechanisms underlying epithelial polarization and polarized cargo transport. In addition, this study can contribute to the understanding of many yet-to-be-discovered pathologies associated with impaired epithelial differentiation,

polarization, and integrity and could therefore serve as a powerful resource for the investigation and characterization of congenital diseases (*Thiagarajah et al., 2018*; *Berni Canani et al., 2010*).

## Materials and methods
### Antibodies and reagents
The following antibodies were commercially obtained and used as stated:

Aminopeptidase N (IF 1:50, #HBB3-153-63, DSHB), Anoctamin 8 (WB 1:500, #HPA049206, Atlas Antibodies), ARHGAP33 (Western blotting [WB] 1:500, #HPA030117, Atlas Antibodies), beta-Actin (WB 1:2000, #A2228, Sigma-Aldrich), cathepsin D (EM 1:50, #219361, Calbiochem), CD63/Lamp3 (EM 1:20, #M1544, Sanquin), Claudin 3 (IF 1:200, #SAB4500435, Sigma-Aldrich), DNP (EM 1:6-1:30, #71-3500, Invitrogen), DPP4 (IF/FACS 1:100, #HBB3/775/42, DSHB; EM 1:10, #AF1180, R&D Systems), E-cadherin (IF 1:200, EM 1:30, #610181, BD Biosciences), GALNT2 (WB 1:1000, #AF7507, Novus Biologicals), Lamp1 (IF 1:200, EM 1:10, #1D4B, DSHB), MTMR2 (WB 1:500, #sc-365184, Santa Cruz Biotechnology), PIP5K1C (WB 1:1000, #3296S, Cell Signaling Technology), Syntaxin3 (IF 1:100, EM 1:10, #133750, Abcam), sucrase-isomaltase (IF 1:100, #HBB2/614/88, DSHB), TM9SF4 (WB 1:1000, #sc-374473, Santa Cruz Biotechnology), TMEM110 (WB 1:1000, #NBP1-69238, Novus Biologicals), and ZO-1 (EM 1:50, #61-7300 Zymed).

Secondary horseradish peroxidase-coupled (HRP) goat anti-mouse and goat anti-rabbit (1:5000, Sigma-Aldrich) were used for WB, and secondary Alexa Fluor-conjugated (Alexa Fluor 488 and 568) goat anti-mouse (1:1000, Life Technologies) and goat anti-rabbit (1:1000, Life Technologies) were used for IF labeling. For labeling of actin filaments, we used phalloidin–Alexa Fluor 568 (1:500, Life Technologies) and for nuclear staining we used Hoechst 3342 (1:10,000, Thermo Fisher Scientific). Secondary antibodies conjugated to 5, 6, 10, or 15 nm colloidal gold particles diluted to 1:50–150 for EM were from British Biocell Intl. and Aurion.

### Plasmids and lentivirus production
For CRISPR/Cas9-mediated depletion, guide RNA (gRNA) targeting sequences for ARHGAP33 (5′-TCCACCGGTGCATATTTGAC-3′ and 5′-GGCACCCTATGAGGGGTACG-3′), TM9SF4 (5′-GCCCAGCAAGATAACCTACA-3′ and 5′-CATCCTTTACTATCATCGGG-3′), Pip5k1c (5′- GAAGTTGGGCCATCGAGGTG-3′ and 5′-GCTCACCCCATCGTGGACGA-3′), Mtmr2 (5′- AGTCGAGGTGAAAATTCTTA-3′ and 5′-GAGGCCGTATCCCAGTAAGT-3′), Tmem110 (5′-GAGCAAGGTCCGCTACCGGA-3′ and 5′-CCGTCAGCGTCCTGGTAGAG-3′), ANO8 (5′-CCGATGACCACACGCTGCTA-3′ and 5′-GAGCACTACCACCGACACGA-3′), and Galnt2 (5′-ACTGCGAGTGTAATGAGCAC-3′ and 5′-GTCGGCCCTACTCAGGACCG-3′) were selected from the Geckov2-CRISPR library, according to their targeting efficiency in the primary CRISPR screen (CRISPR Design; Zhang lab, Massachusetts Institute of Technology; *Hsu et al., 2013*). gRNAs were cloned into a lentiCRISPRv2 vector via BsmBI restriction enzyme sites. lenti-CRISPRv2 was a gift from F. Zhang (Massachusetts Institute of Technology, Cambridge, MA; Addgene plasmid 52961; *Sanjana et al., 2014*). For the generation of knockout cell lines, lentiviral plasmids were cotransfected using Lipofectamine LTX (Invitrogen) transfection reagent together with pVSV-G and psPAX2 in the Hek293LTV producer cell line. Virus containing supernatants were collected after 48 and 72 hr after transfection and directly used for CaCo2 cell infection. 6 days after infection, cells were selected with 10 µg/ml puromycin (Sigma-Aldrich) or 20 µg/ml blasticidin S (Invitrogen). Depletion efficiency was verified via WB.

### Cell culture
Hek293LTV, CaCo2 WT (ATCC, HTP-37), and KI cells were cultured in DMEM (Sigma-Aldrich) containing high glucose, sodium pyruvate, 100 U/ml penicillin (Sigma-Aldrich), 100 µg/ml streptomycin (Sigma-Aldrich), 5% nonessential amino acids (Gibco), and 10% FBS (Gibco) in a humidified atmosphere with 5% $CO_2$ at 37°C. For experiments requiring fully polarized growth conditions, CaCo2 cells were seeded on 24 mm or 75 mm filters (Costar Transwell; pore size of 0.4 µm; Corning) and cultured for 14–28 days. For 3D cyst assays, CaCo2 cells were cultivated and processed as described previously (*Vogel et al., 2015b*; *Jaffe et al., 2008*). For this purpose, $5 \times 10^4$/mL single cells were embedded in Matrigel (BD Biosiences, #356231), plated on 8× chamber slides (Lab-Tek-chamber slide, Sigma)

chamber slides (10,000 cells/slide) and grown for 7 days (*Román-Fernández et al., 2018*). All cell lines were regularly tested negative for mycoplasm.

## TEER measurements

Transepithelial electric resistance (TEER) measurements were performed in CaCo2 wildtype and the generated KO cell lines. TEER was measured using an STX2 electrode together with the EVOM epithelial volt-ohmmeter from World precision instruments. TEER measurements were performed on days 0, 4,7, 11, 14, and 19 after cells were seeded on transwell filters at confluence for polarization. The measurements were performed at three different areas on the filter inserts and calculated as described previously (*Klee et al., 2020*).

## Genome-wide CRISPR screen

For the CRISPR screen in polarized CaCo2 cells, we used the 2-vector system (lenti-guide Puro; Addgene #1000000049; Feng Zhang Lab, *Sanjana et al., 2014*). For the generation of the target CaCo2 cell line, we introduced the vector encoding Cas9 (lentiCas9-Blast) for stable expression with a lentivirus to CaCo2 cells.

For virus production containing the sgRNA library, HEK293T cells seeded to 150 mm dishes were transfected with 21 µg of the human gRNA pooled library in lentiGuide-Puro (Addgene #1000000049), 15.75 µg of pSPAX2 and 5.25 µg of pVSV-G plasmids. Viral supernatants were collected after 36 hr and 50 hr and concentrated with Amicon ultra-15 centrifugation tubes (Merck). For storage and further usage, samples were snap-frozen in liquid nitrogen.

The CRISPR/Cas9 screen was performed as described in previous studies by *Hutter et al., 2020*; *Shalem et al., 2014*. Two replicates were performed. For each replicate, $2 \times 10^8$ CaCo2-Cas9 screen cells were transduced with a virus preparation containing 16 µg/ml polybrene at an MOI of 0.2 and seeded at a low density ($10^7$ cells/150 mm plate) to 20 150 mm culture dishes. 72 hr after infection, selection with 10 µg puromycin was started and cells were selected for 7 days. After 7 days of selection, surviving screen cells were pooled and evenly distributed to 100 mm culture dishes to obtain confluent monolayers. The cells were then cultivated for polarization as confluent monolayers for 18 days. After 18 days, cells were detached with StemPro Accutase Cell Dissociation Reagent (Thermo Fisher Scientific, #A1110501) and stained for FACS. Thereby, cell suspensions were washed after detachment 2× with ice-cold PBS and subsequently incubated in PBS containing 1% FBS and anti-DPP4 antibody (1:100) on ice for 20 min. After incubation, cells were washed 2× in ice-cold PBS and then incubated with PBS containing 1% FBS and a secondary anti-mouse Alexa Fluor-488 antibody (1:1000) on ice for 20 min. After the incubation, cells were washed 2× in ice-cold PBS, resuspended in PBS containing 2% FBS, and subjected to FACS sorting using an ARIA III (Becton Dickinson). For each replicate, approximately 1.5 Mio cells corresponding to the lowest 10% Alexa-488 (A-488-negative) of the total cell population was sorted. For colchicine experiments, CaCo2 WT cells were treated with 20 µM colchicine (Merck, #C9754) for 2.5 hr prior to FACS sorting. Unsorted cells were saved and used as control. Genomic DNA (gDNA) was isolated from sorted and unsorted cells using the Nucleospin Tissue Mini Kit (Macherey-Nagel) and sgRNA sequences were retrieved by a nested PCR approach that pre-amplified sgRNAs in a first round with primers specific to the lentiGuide-Puro construct (5′-AGAGGGCCTATTTCCCATGA-3′) and added stagger bases, specific barcodes, and the Illumina adapters in the second round. The PCR products were separated on a 1% agarose gel, purified by gel extraction, quantified, and then pooled before sequencing on a Hiseq4000 (Illumina) in collaboration with the Biomedical Sequencing Facility (BSF, Vienna, Austria). Sequencing data were analyzed with the publicly available online tools GenePattern (*Chapman et al., 2006*) and Galaxy (*Afgan et al., 2018*). Reads were first demultiplexed and trimmed followed by alignment of the sgRNA sequences to a reference using Bowtie2 (*Langmead and Salzberg, 2012*). SgRNAs enriched in the sorted A-488-negative populations were identified using the edgeR shRNaseq tool (Table CRISPR screen Analysis for Enriched sgRNAs in the A-488-negative; *Dai et al., 2014*).

## Immunoblots

Total cell lysates were prepared and Western blot analysis was performed as described previously (*Cattelani et al., 2021*). Cells were washed in 1× cold PBS, scraped from respective culture plates and pelleted with $1500 \times g$ for 5 min at 4°C. Cell pellets were resuspended in lysis buffer (50 mM HEPES

pH 8.0, 150 mM NaCl, 5 mM EDTA pH 8.0, 0.5% NP-40, 50 mM NaF, 10 µg/ml leupeptin, 0.4 mM pefablock, 1 µ/ml pepstatin, 10 µg/ml aprotinin, 0.5 mM PMSF, 1 mM $N_3VO_4$) and lysed for 60 min on ice. Then, lysates were centrifuged at 13.000 × g for 15 min and cleared lysate was obtained. Lysates were separated by SDS-polyacrylamide gel electrophoresis (PAGE). Polyacrylamide gels were prepared consisting of stacking (125 mM Tris pH 6.8, 4% acrylamide/bis solution [37:5:1], 6% glycerol, 0.1% SDS, 0.075% APS, and 0.1% TEMED) and resolving gels (0.375 mM Tris pH 8.8, 7–15% acrylamide/bis solution [37:5:1], 0.1% SDS, 0.05% APS, and 0.05% TEMED). All SDS-PAGE gels were run in 192 mM glycine, 25 mM Trisma Base, and 0.1% SDS. After separation, the proteins were wet transferred onto 'Amersham Protran 0.2 µm NC' nitrocellulose membranes (GE10600002, Sigma-Aldrich, Handels Gmbh, Vienna, Austria) at constant 80 V for 1,5 hr. The wet transfer buffer contained 25 mM Tris, 192 mM glycine, 0.1% SDS, and 20% methanol (vol/vol), adjusted to pH 8.3. Membranes were subsequently blocked in 3% BSA (fraction V), 1 mM EDTA, 0.05% Tween20, and 0.02% $NaN_3$, and probed with the respective antibodies.

## Immunofluorescence microscopy

Immunofluorescence stainings on cells grown and polarized on glass coverslips or of 3D cyst cultures were performed as described previously (*Vogel et al., 2015b*; *Vogel et al., 2017b*). Briefly, for stainings of polarized 2D monolayers, cells grown on glass coverslips were fixed with 4% formaldehyde (made from paraformaldehyde) at room temperature for 3 hr or 100% methanol at –20°C for 5 min, respectively. Cells stained with anti-DPP4, anti-stx3, anti-HA, and phalloidin were fixed with formaldehyde, while for stainings with anti-ZO-1, anti-ECadherin, anti-clau3, anti-NaK-ATPase, and anti-Moesin, cells were fixed with methanol. CaCo2 cysts were prepared for IF microscopy as described previously (*Jaffe et al., 2008*). Confocal stacks from monolayers/cysts mounted in Mowiol were taken on confocal fluorescence microscopes (SP5 and SP8; Leica) using a glycerol 63× lense with a numerical aperture of 1.3 (Leica) on a Leica SP5 microscope and a glycerol 93× lense with a numerical aperture of 1.3 on the Leica SP8 microscope at room temperature. As recording software, LASAF 2.7.3. (Leica) and LAS X (Leica) were used. All images were deconvolved using Huygens Professional Devonvolution and Analysis Software (Scientific Volume Imaging) and exported using Imaris 3D rendering (Bitplane) and finally adjusted for brightness and contrast using Fiji ImageJ software.

## Electron microscopy

TEM for morphology of filter-grown, polarized monolayers included rapid cryo-immobilization through means of high-pressure freezing, followed by freeze-substitution and epoxy resin embedding as described previously (*Vogel et al., 2015b*; *Ruemmele et al., 2010*). Immunogold EM was described previously; in brief, polarized monolayers grown for 14 days in Petri dishes were fixed with 4% formaldehyde or 4% formaldehyde plus 0.1% glutaraldehyde and processed for Tokuyasu-ultracryotomy (*Vogel et al., 2017a*; *Tokuyasu, 1973*). Optionally, cells were subjected to serum-stravation overnight, followed by incubation for 2 hr with DAMP (*Orci et al., 1986*; 3-(2,4-dinitroanilino)–3'amino-N-methyl propyl-amine, #D1552 from Molecular Probes; 30 µM/l dissolved in serum-free medium) prior to aldehyde fixation. Analysis of thin sections was performed with a Philips CM120 (now Thermo Fisher Scientific), equipped with a MORADA digital camera and iTEM software (EMSIS, Münster, Germany). Image contrast, brightness, greyscale, and sharpness were adjusted with Photoshop CS6 (Adobe, San José, CA). For morphometry as previously described (*Vogel et al., 2015a*; *Hess and Huber, 2021*), we used samples from ≥2 independent cell culture experiments (i.e., n ≥ 2, with several technical replicates each; 40–120 endo/lysosomes per condition, ≥40 cells/condition for ectopic microvilli). Morphometry was carried out on digital micrographs taken at primary magnifications of ×7100 to ×15000 by using measurement tools of the iTEM software.

SEM of filter-grown polarized monolayers was performed with a DSM 982 Gemini (ZEISS, Oberkochen, Germany) as described previously (*Ruemmele et al., 2010*); briefly, sample processing included chemical fixation, dehydration, critical point drying and sputter coating.

## GO term analysis

For manual GO term analysis, each of the 89 significantly enriched genes from out CRISPR screen were subjected to a manual GO term search using https://www.uniprot.org/ and https://www.ebi.ac.uk/QuickGO/annotations. For each gene, three GO terms were listed, prioritizing most common

GO terms suggested by the QuickGO-Database and GO terms indicating a relation to the secretory pathway, for each of the three categories, biological process (BP), molecular function (MF), and cellular compartment (CC). According to commonalities in the individual sets of GO terms, genes were then grouped and graphically visualized using Affinity Designer.

## Statistics and software

The software used, if not already specified, were Affinity Designer (version 1.9.3), Fiji/ImageJ (version 2.1.0/1.53c), GraphPad Prism 9 (version 9.1.0), and Serial Cloner (version 2.6); for the analysis and visualization of FACS-Data, we used FlowJo (Becton Dickinson). Dot box plot graphs were generated and the unpaired Mann–Whitney U test was calculated using R (R Core Team (2021). R: A language and environment for statistical Computing. R Foundation for Statistical Computing, Vienna, Austria https://www.R-project.org) and ggplot2 package.

## Acknowledgements

We thank the Biomedical Sequencing Facility (BSF) at the Research Center for Molecular Medicine (CeMM) of the Austrian Academy of Science (ÖAW) in Vienna for their next-generation-sequencing (NGS) services, especially Jan Laine and Michael Schuster.

We thank Karin Gutleben, Barbara Witting, and Angelika Flörl (Institute of Histology and Embryology, Medical University of Innsbruck), and Caroline Krebiehl (Institute of Cell Biology, Medical University of Innsbruck) for technical assistance.

We further thank the Austrian Academy of Science (ÖAW) for grating the DOC-Scholarship to Katharina MC Klee and thereby supporting this work. This work was supported by the Austrian Science Fund (FWF P35805) and the Jubiläumsfonds der Österreichischen Nationalbank (grant no. 18019).

## Additional information

### Funding

| Funder | Grant reference number | Author |
|---|---|---|
| Austrian Science Fund | P35805-B | Georg F Vogel |
| Austrian Academy of Sciences | DOC-Scholarship | Katharina MC Klee |
| Jubiläumsfonds der Österreichischen Nationalbank | 18019 | Georg F Vogel |

The funders had no role in study design, data collection and interpretation, or the decision to submit the work for publication.

### Author contributions

Katharina MC Klee, Conceptualization, Funding acquisition, Investigation, Visualization, Methodology, Writing - original draft, Writing – review and editing; Michael W Hess, Conceptualization, Investigation, Visualization, Methodology, Writing - original draft, Writing – review and editing; Michael Lohmüller, Sebastian Herzog, Kristian Pfaller, Investigation, Methodology; Thomas Müller, Writing – review and editing; Georg F Vogel, Conceptualization, Supervision, Funding acquisition, Investigation, Visualization, Writing - original draft, Project administration, Writing – review and editing; Lukas A Huber, Conceptualization, Funding acquisition, Project administration, Writing – review and editing

### Author ORCIDs

Michael W Hess  http://orcid.org/0000-0002-5154-3553
Michael Lohmüller  http://orcid.org/0000-0002-7712-3143
Georg F Vogel  http://orcid.org/0000-0002-2515-4490
Lukas A Huber  http://orcid.org/0000-0003-1116-2120

Decision letter and Author response
Decision letter https://doi.org/10.7554/eLife.80135.sa1
Author response https://doi.org/10.7554/eLife.80135.sa2

## Additional files

### Supplementary files

• Supplementary file 1. Raw screen sequencing data and GO analyses.

• Supplementary file 2. Synopsis of phenotype characteristics of CaCo2-WT versus TM9SF4-, ANO8-, PIP5K1C-, ARHGAP333-, MTMR2-, and TMEM110-KO cells.

• Supplementary file 3. Phenotype quantification of CaCo2-WT versus TM9SF4-, ANO8-, PIP5K1C-, ARHGAP333-, MTMR2-, and TMEM110-KO cells. ELY, endolysosomes; LY, lysosomes; MVI, microvillus inclusions; latMV, lateral microvillar assemblies; par MV, paracellular microvillar clusters.

• MDAR checklist

### Data availability

Next generation sequencing data was made available in Dryad https://doi.org/10.5061/dryad.m0cfxpp62. Source Data files have been provided for Figure 3.

The following dataset was generated:

| Author(s) | Year | Dataset title | Dataset URL | Database and Identifier |
|---|---|---|---|---|
| Vogel GF | 2023 | NGS raw sgRNA-counts | https://dx.doi.org/10.5061/dryad.m0cfxpp62 | Dryad Digital Repository, 10.5061/dryad.m0cfxpp62 |

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
