## [Editor Report]

In this work, Klee et al. carried out a genome-wide CRISPR/Cas9-based screen in human intestinal cell line CaCo2 to uncover factors regulating apical localization of a brush border enzyme. Their findings identified dozens of genes and characterized novel players in apical membrane transport including TM9SF4, anocatmin 8, and ARGAP33. This work provides a useful resource for the study of apical polarity and may aid in the understanding of digestive diseases.

---

## [Decision Letter]

**Decision letter after peer review:**

Thank you for submitting your article "A CRISPR-screen in intestinal epithelial cells identifies novel factors for polarity and apical transport" for consideration by *eLife*. Your article has been reviewed by 3 peer reviewers, and the evaluation has been overseen by a Reviewing Editor and Suzanne Pfeffer as the Senior Editor. The following individuals involved in the review of your submission have agreed to reveal their identity: Fernando Martin-Belmonte (Reviewer #2); David M Bryant (Reviewer #3).

Essential revisions:

1. Previous work by other researchers using cell models similar to those described in this work (i.e., MDCK cells) has shown that apical membrane proteins are internalized in vesicles called vesicular apical compartment (VAC) when detached from the substrate. It would be necessary for researchers to demonstrate that this process is not affecting their apical transport screening by using the appropriate controls.

2. The authors should use rescue systems and/or additional gRNAS to demonstrate that the identified phenotypes are not the product of off-target effects of the gRNAs used in the screening and later in the generation of KO cells.

3. More quantitative phenotypic analysis of the 7 selected target genes should be included. The authors should describe penetrance/frequency and severity of the phenotypes described (inclusions, mislocalized proteins, etc). and add quantification and graphical representation of different lysosome sizes in WT and all KD cell lines.

4. Loss of apical sorting/trafficking machinery results in impaired localization of specific classes of membrane proteins (for example, O- vs N-glycosylated cargoes). This study would therefore be strengthened by analysis of an additional apical membrane protein marker for some of the target gene KD cell lines.

---

## [Author Response]

Essential revisions:1. Previous work by other researchers using cell models similar to those described in this work (i.e., MDCK cells) has shown that apical membrane proteins are internalized in vesicles called vesicular apical compartment (VAC) when detached from the substrate. It would be necessary for researchers to demonstrate that this process is not affecting their apical transport screening by using the appropriate controls.

We agree that this valid criticism needs experimental control. Therefore, we have treated filter grown polarized CaCo2 WT cells with 20 µM of colchicine for 2.5 hours (according to Gilbert and Rodriguez-Boulan) and repeated the FACS assay used for phenotype screening. The surface staining intensity of DPP4 was not altered upon colchicine treatment compared to untreated WT cells. Therefore, we conclude that the potential formation of vacuolar apical compartments under this condition would not influence the results from the initial FACS screening assay. We have incorporated the experimental data in Figure 1 —figure supplement 1 A and B.

2. The authors should use rescue systems and/or additional gRNAS to demonstrate that the identified phenotypes are not the product of off-target effects of the gRNAs used in the screening and later in the generation of KO cells.

We agree with the reviewers’ comment and have generated KO cells with an additional set of gRNAs. We have included these KOs into our phenotypic analyses by immunofluorescence microscopy and corresponding quantifications.

3. More quantitative phenotypic analysis of the 7 selected target genes should be included. The authors should describe penetrance/frequency and severity of the phenotypes described (inclusions, mislocalized proteins, etc). and add quantification and graphical representation of different lysosome sizes in WT and all KD cell lines.

We thank for this valuable suggestion and have added a detailed quantification by immunofluorescence and electron microscopy of the respective phenotypes to the corresponding figures and in Supplementary File 3. On the basis of those new data we were also able to improve the verbal description of organelle size and frequency in the synoptic table (Supplementary File 2).

4. Loss of apical sorting/trafficking machinery results in impaired localization of specific classes of membrane proteins (for example, O- vs N-glycosylated cargoes). This study would therefore be strengthened by analysis of an additional apical membrane protein marker for some of the target gene KD cell lines.

We thank the reviewers for this suggestion and have extended the analyses to the apical membrane proteins aminopeptidase N (APN) and sucrase-isomaltase (SI). Interestingly, we have found APN delocalized to actin-rich intracellular compartments in ARHGAP33-KO cells, whereas the apical localization of SI remained unaffected in all KO cells. We have included this data in Figure 8 —figure supplement 2 and 3.